# Telomere-to-telomere assemblies of 142 strains characterize the genome structural landscape in *Saccharomyces cerevisiae*

**Samuel O'Donnell** [1,7], **Jia-Xing Yue** [2,3,7], **Omar Abou Saada**[4], **Nicolas Agier**[1], **Claudia Caradec**[4], **Thomas Cokelaer** [5,6], **Matteo De Chiara**[3], **Stéphane Delmas**[1], **Fabien Dutreux**[4], **Téo Fournier**[4], **Anne Friedrich** [4], **Etienne Kornobis** [5,6], **Jing Li** [2,3], **Zepu Miao**[2], **Lorenzo Tattini** [3], **Joseph Schacherer** [4,8 ✉], **Gianni Liti** [3,8 ✉] **& Gilles Fischer** [1,8 ✉]

Pangenomes provide access to an accurate representation of the genetic diversity of species, both in terms of sequence polymorphisms and structural variants (SVs). Here we generated the *Saccharomyces cerevisiae* Reference Assembly Panel (ScRAP) comprising reference-quality genomes for 142 strains representing the species' phylogenetic and ecological diversity. The ScRAP includes phased haplotype assemblies for several heterozygous diploid and polyploid isolates. We identified circa (ca.) 4,800 nonredundant SVs that provide a broad view of the genomic diversity, including the dynamics of telomere length and transposable elements. We uncovered frequent cases of complex aneuploidies where large chromosomes underwent large deletions and translocations. We found that SVs can impact gene expression near the breakpoints and substantially contribute to gene repertoire evolution. We also discovered that horizontally acquired regions insert at chromosome ends and can generate new telomeres. Overall, the ScRAP demonstrates the benefit of a pangenome in understanding genome evolution at population scale.

Single-molecule long-read sequencing provides access to gapless genome assemblies, including repetitive chromosomal regions that generally remain unassembled with previous technologies. This is best exemplified in the rapid increase in contiguity of the human genome[1], especially thanks to ultra-long reads from Oxford Nanopore Technology (ONT)[2]. Recently, the telomere-to-telomere (T2T) consortium released the first complete 'T2T' assembly of two human chromosomes[3–5], followed by the release of the first gapless human genome, including nearly 200-Mb new sequences[6]. Complex plant genomes and classical model organisms have also seen improvements in assembly contiguity, thanks to long-read technologies[7–11].

These advancements allowed few species to have multiple reference-like contiguous genomes, which include model organisms and species of anthropocentric importance such as *Escherichia coli*[12],

[1]Sorbonne Université, CNRS, Institut de Biologie Paris-Seine, Laboratory of Computational and Quantitative Biology, Paris, France. [2]State Key Laboratory of Oncology in South China, Collaborative Innovation Center for Cancer Medicine, Guangdong Key Laboratory of Nasopharyngeal Carcinoma Diagnosis and Therapy, Sun Yat-sen University Cancer Center, Guangzhou, China. [3]Université Côte d'Azur, CNRS, INSERM, IRCAN, Nice, France. [4]Université de Strasbourg, CNRS, GMGM UMR 7156, Strasbourg, France. [5]Biomics Technological Platform, Center for Technological Resources and Research (C2RT), Institut Pasteur, Paris, France. [6]Bioinformatics and Biostatistics Hub, Computational Biology Department, Institut Pasteur, Paris, France. [7]These authors contributed equally: Samuel O'Donnell, Jia-Xing Yue. [8]These authors jointly supervised this work: Joseph Schacherer, Gianni Liti, Gilles Fischer. ✉e-mail: joseph.schacherer@unistra.fr; gianni.liti@unice.fr; gilles.fischer@sorbonne-universite.fr

*Drosophila melanogaster*[10,13], *Solanum lycopersicum*[14], *Glycine max*[15], *Oryza sativa*[8,16], *Bombyx mori*[17] and humans[18–20]. The baker's yeast, *Saccharomyces cerevisiae*, has in total 68 long-read genome assemblies of nonreference strains[21–30]. These data have been used to quantify contiguity improvements over short-read data[25], create genome-wide maps of transposable elements (TEs)[22,24,25], characterize subtelomeric regions[29], phase haplotypes and detect large structural variants (SVs)[22,25,26,29,30]. However, the contiguity of the available genome assemblies varies widely in *S. cerevisiae* and only a small subset of them reached the chromosome-level contiguity. Furthermore, the sampling remains limited with many clades lacking a representative reference genome and no polyploid genomes have been included despite their abundance (11.5% of the isolates)[31]. Lastly, phasing haplotypes of diploid and polyploid genomes is challenging, preventing haplotype inference and measures of heterozygosity.

Here we generated the *S. cerevisiae* Reference Assembly Panel (ScRAP) comprising T2T genome assemblies for 142 isolates that sample the species genomic space. The quality of these genomes exceeds the reference gold standard and allows us to precisely characterize SVs and complex regions at a scale that has not yet been achieved in other species.

## Results

### The ScRAP

The ScRAP includes 142 strains that cover the species geographical and ecological distributions and its ploidy and heterozygosity levels (Fig. 1a,b and Supplementary Table 1). The panel comprises 197 nuclear and 136 mitochondrial genome assemblies, including 100 newly sequenced genomes, among which haplotype-resolved assemblies are available for both diploid and polyploid genomes (Table 1 and Supplementary Tables 1–3). Genomic metrics reveal high contiguity and completeness levels across all assemblies (Supplementary Note 1). The ScRAP provides reference-quality genomes across all main phylogenetic clades[31,32] (Fig. 1c and Supplementary Note 2). T2T haplotype-resolved diploid assemblies show that sister haplotypes (HPs; haplotype 1 (HP1) and haplotype 2 (HP2)) always grouped together in the tree and shared the same admixture profile (Fig. 1c,d). The most striking difference was observed between the two HPs of the Wine/European MC9 (AIS) strain for which the branch length of HP2 (AIS_HP2) is disproportionately longer compared to all other terminal branches (Fig. 1c), which is driven by chromosome-scale introgressions of chromosomes VI and VII from a highly diverged species (see Whole-chromosome introgressions).

We applied a strict molecular clock model to time the major founding events of the species history (Methods). Consistent with previous estimates, *S. cerevisiae* has split from its sister species *Saccharomyces paradoxus* 5.7-1.7 MYA (Supplementary Table 4). The first split of the most diverged lineage (CHN-IX/TW1) occurred 680-180 KYA. The species' origin was followed by a single out-of-China event that founded the rest of the world population 290-80 KYA. The Wine/European lineage separated 55-15 KYA from the wild Mediterranean oak population, which likely represents its wild ancestor[33].

### The species structural diversity

We identified a total of 36,459 SVs by pairwise whole-genome alignments against the S288C reference genome (Fig. 2a; Methods). These calls consist of copy number variants (CNVs) >50 bp, including deletions, insertions, duplications and contractions of repetitive sequences and copy-neutral rearrangements including inversions (>1 kb) and translocations (>10 kb). They originated from 4,809 nonredundant large-scale rearrangements that are shared at varying frequencies across the 141 nonreference strains (Table 1 and Supplementary Table 5). This nonredundant SV catalog covers ca. 80% of the estimated whole species structural diversity that we predicted to contain approximately 6,000 SVs (Fig. 2b and Table 1).

Phasing heterozygous genomes added a large number of SVs that would have remained undetected using only collapsed assemblies. On average, 33% of calls detected in phased strains were only validated by phased assemblies (Extended Data Fig. 1a) and 53% of them are heterozygous (Table 1 and Extended Data Fig. 1b). Notably, both the proportion of calls validated only in the presence of phased genomes and the proportion of heterozygous variants increases with ploidy. The median number of SVs also increases with ploidy, from 219 SVs in haploids to 453 in tetraploids (Extended Data Fig. 1c). We plotted the number of SVs as a function of the number of SNVs/indels for each strain and observed a positive correlation (Fig. 2c). However, the SV number scales up more rapidly with higher ploidy than that of SNVs (Fig. 2c). Additionally, for a given number of SNVs/indels, the number of SVs is also systematically higher in heterozygous than in homozygous genomes. These observations suggest that SVs preferentially accumulate or are better tolerated in higher ploidy and heterozygous genomes.

There is a median of 240 SVs per strain with a maximum of 639 events in the highly heterozygous tetraploid YS8(E) (BTE) strain (Table 1 and Supplementary Table 6). The number of SVs does not differ between domesticated and wild isolates (Wilcoxon signed-rank test, *P* = 0.53). Deletions and insertions are the most frequent types of SVs (~100 events per strain), followed by duplications and contractions (10–20 events per strain), translocations and inversions are rarer (only a few occurrences per strain; Supplementary Table 6). Most SVs are present at low frequencies in the population, with 34% of the events being found in a single genome and 91% with a minor allele frequency <0.1 (Fig. 2d), suggesting that SVs are mostly deleterious or recent.

All types of SVs, except the inversions, are primarily confined to subtelomeric regions (Fig. 2e), in accordance with the high evolutionary plasticity of these regions[29]. Insertions more frequently contain repetitive sequences (82%) compared to deletions, duplications and inversions (41–47%). The distribution of event sizes, excluding translocations, shows that small SVs are the most prevalent with 58% of the events being <1 kb and only 9% being >10 kb (Fig. 2f). This distribution shows two clear peaks around 300 bp and 6 kb for deletions, insertions and inversions corresponding to solo-long terminal repeats (LTRs) and full-length Ty elements. The mobility of Ty elements directly accounts for 59% of all insertions (1,571 events) and 16% of deletions through inter-LTR recombination (218 events). This unbalance is explained by the limited number of Ty elements in the reference genome that can be interpreted as a deletion when absent from other genomes. Interestingly, 19% and 8% of all duplications and contractions (representing 74 and seven cases, respectively), also resulted from tandem Ty movements. Altogether 39% of all SVs result from the insertion and deletion of Ty elements.

We found a clear enrichment of repetitive sequences (LTRs, Tys, tRNAs, Y' and X elements) at the junction of SVs as well as a substantial underrepresentation of coding DNA sequences (CDS) overlapping with those breakpoints (Extended Data Fig. 2). Interestingly, we found a substantial association between autonomously replicating sequences (ARSs) and SV breakpoints. We extracted all ARSs from ORIdb[34] and showed that the ARS−SV association is greater as the likelihood of the ARS being fired increases (Fig. 2g).

### SVs shape the population gene repertoire

We found that nearly 40% of the SVs (1,876 of 4,809) directly impacted protein-coding genes (Table 1), excluding SVs involved in the insertion and deletion of Ty elements. Interestingly, this proportion drops to 3% for essential genes. The most frequent case is by far the situation where both breakpoints of a given SV lie within the same gene. We found 1,170 such cases of intragenic SVs, mostly corresponding to insertions and to a lesser extent to deletions and duplications (Fig. 2h). Most contractions of repetitive sequences also belong to this category as 78 of 93 fall within

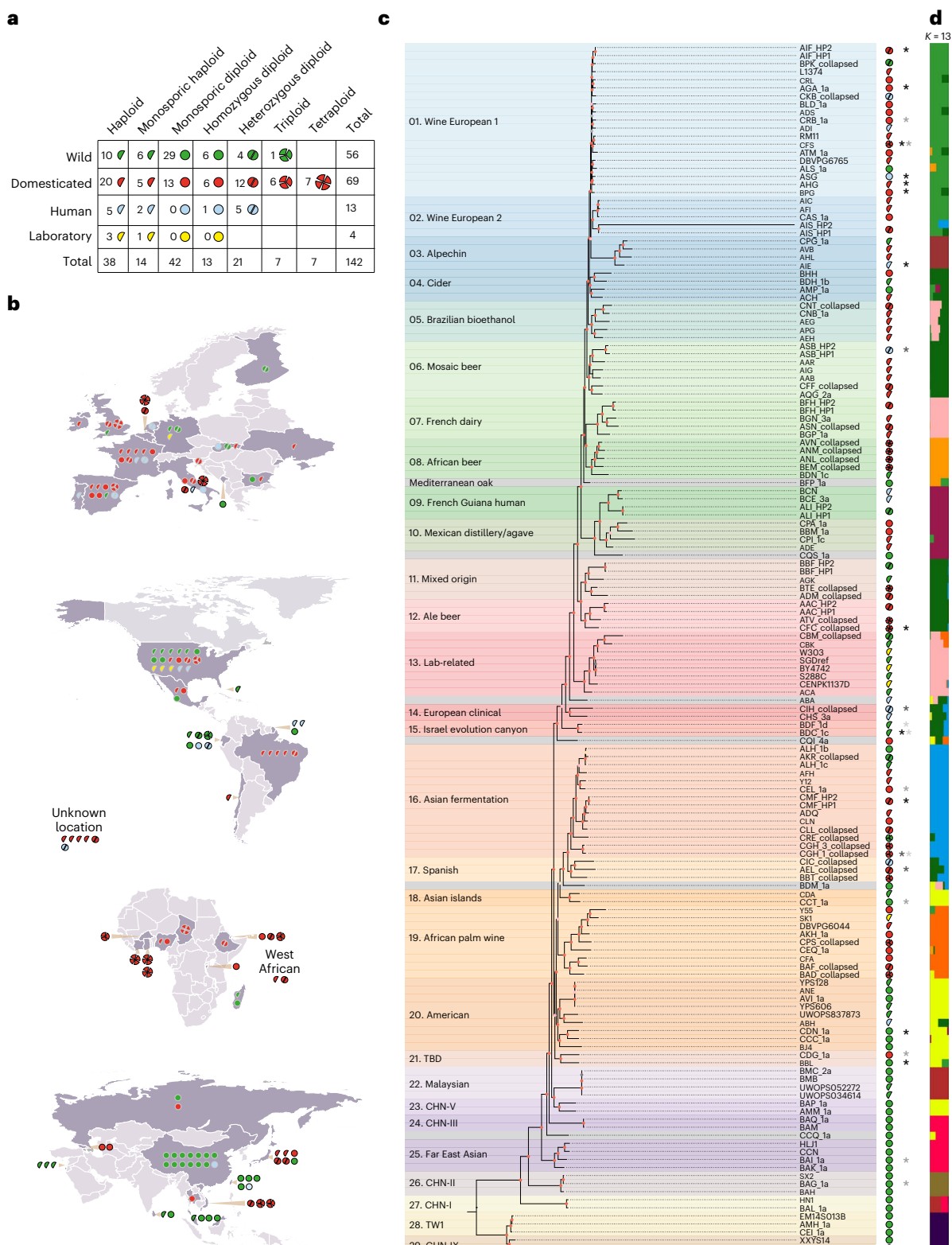

**Fig. 1 | The origin and genomic composition of the 142 ScRAP strains.**
**a**, Ecological origin, ploidy and zygosity description of the 142 ScRAP strains.
Colors are used as keys to symbolize the strain origin (wild (green), domesticated
(red), human (blue) or laboratory (yellow)) and shapes symbolize their ploidy
and zygosity levels (haploid (one-slice half circle), homozygous diploid
(full circle), heterozygous diploid (two-sliced circle), heterozygous triploid and
tetraploid (three- and four-sliced circle). The haploid category contains both
natural and engineered (Δho) strains. All triploid and tetraploid strains are
heterozygous except for the homozygous triploid strain isolated in the USA.
**b**, Geographical origin of the isolates. The shape and colors of the symbols are

as in **a**. **c**, Phylogenetic tree based on the concatenated protein sequence
alignment of 1,612 1:1 orthologs. The tree was rooted by including 23 strains
from other *Saccharomyces* species (not presented in the figure). The symbols
on the right recall the ecological origin, ploidy and zygosity of all isolates, as
described in **a**. The presence of aneuploid chromosomes is labeled by an asterisk
with varying gray levels discriminating between several cases relative to the
1,011 genome survey[31]—black, previously detected; dark gray, not previously
detected; middle gray, previously absent and newly gained; light gray, previously
present but newly lost. **d**, Genetic ancestry of the population as defined by
running ADMIXTURE with *k* = 13.

## Table 1 | Key figures of the ScRAP and associated SVs

| | Key figures |
|---|---|
| ScRAP | 142 strains: 41 haploids, 56 monosporic isolates, ten homozygous diploids, 21 heterozygous diploids, one homozygous triploid, six heterozygous triploids and seven heterozygous tetraploids |
| | 197 nuclear assemblies: 142 haploid/collapsed assemblies, 2×21 haplotype diploid assemblies, six phased triploid assemblies and seven phased tetraploid assemblies |
| | 136 mitochondrial assemblies |
| SVs | Total structural diversity in *S. cerevisiae*: ~6,000 independent large-scale rearrangements |
| | ScRAP structural diversity: 4,809 independent large-scale rearrangements generating 36,459 SVs |
| | Median of 240 SVs per strain |
| | 34% of SVs found in a single isolate |
| | 53% of SV calls detected in phased strains are heterozygous |
| | 39% of all SVs result from the insertion and deletion of Ty elements |
| | 40% of SVs directly impacted protein-coding genes |
| | 4.4% of SV breakpoints impact nearby gene expression in complete medium |
| | Up to 18% of aneuploid chromosomes are complex, that is, associated with large SVs |

coding sequences. It is difficult to predict the functional outcome of intragenic SVs as each event can disrupt, or not, its corresponding coding sequence depending on its size and position relative to the reading frame. We found 508 SVs where at least one entire gene lies between the two breakpoints, corresponding to 345 deletions, 84 inversions and 27 duplications containing on average five, 30 and two genes, respectively. In total, the 345 deletions comprised 525 different genes that have been fully deleted in at least one HP. The two last categories, gene disruption and gene fusion, comprise all SVs for which either one or both breakpoints lie within a protein-coding gene. Note that these two categories are not mutually exclusive with the previous one as a given event can both contain entire genes and disrupt or fuse with other genes at its breakpoints. We identified 450 cases of gene disruptions that produce gene truncations by merging the internal part of a gene with an intergenic region. We also found 145 putative gene fusion events where both breakpoints of a given SV fall within different genes. These events can create new chimeric genes, although they likely comprise undetermined proportions of both in-frame and out-of-frame fusions. Surprisingly, about half of the translocations (98 of 200) resulted in gene disruption (*n* = 71) or fusion (*n* = 27) at their breakpoints, in contrast to the general assumption that translocations occur mostly between TEs. Altogether, we identified 1,698 complete gene deletions and duplications, as well as 1,513 gene structure alterations at the origin of new gene sequences that can substantially expand the gene repertoire of the species.

## SVs can impact gene expression near breakpoints

SVs can influence gene expression by impacting the sequence of the open reading frame, modifying their copy numbers or changing their regulatory elements. By leveraging a recent survey that generated the transcriptome of more than 1,000 *S. cerevisiae* isolates[35], we explored the relationship between gene expression and SVs. For 51 isolates, we analyzed the expression levels for 6,445 transcripts against 1,876 SVs, encompassing a similar proportion of the different SV types than the entire dataset (Extended Data Fig. 3a). We first defined a set of 2,808 SV–gene pairs with more than half of the pairs involving insertion or deletion events. We then compared the expression of genes associated or not with a given SV (Extended Data Fig. 3b). We found that 124 SV–gene pairs (4.4%; Supplementary Table 7), encompassing 97 unique SVs, showed substantial differential expression changes (Table 1). This impact seems to be subtle, but the transcriptomic data were obtained from a single condition (rich medium) and we restricted the analysis only to direct *cis* effects. Interestingly, this proportion is variable depending on the type of SVs (Fig. 2i) with more than 5% of pairs involving deletions and duplications, and only 1% of pairs involving inversions, translocations and contractions.

We explored the difference between SVs located in coding and noncoding regions, restricting to insertions and deletions events. In total, 7.3% of SV–gene pairs (60 of 815) affecting coding sequences are associated with substantial differences in expression, mainly by reducing or suppressing expression (Fig. 2i). By contrast, only 3.1% (23 of 726) of pairs present in noncoding regions were detected as substantially impacting gene expression. Overall, these results demonstrate a varying impact on gene expression depending on the SV type and location.

## SVs produce complex aneuploid chromosomes

We identified 26 whole-chromosome aneuploidies affecting 18 of the 142 isolates (Supplementary Table 8). Interestingly, we also discovered a complex type of aneuploidies that comprise large SVs such as translocations, horizontal gene transfer (HGT) insertions and large (~100 kb) deletions (Supplementary Table 8). We identified eight complex aneuploidies in seven strains, which represents 24% of all aneuploidies in the ScRAP. We fully resolved the chromosomal organization in five strains (Fig. 3a) and confirmed that all seven complex aneuploidies were already present when the strains were initially Illumina sequenced[31]. We reanalyzed 993 strains (84 from ref. 36 and 909 from ref. 31) to detect both simple and complex aneuploidies. We found that a large proportion of aneuploid chromosomes (up to 18%) are associated with large SVs at the population scale (Supplementary Note 3 and Supplementary Tables 9 and 10). Interestingly, we found that complex aneuploidies involve larger chromosomes as compared to simple aneuploidies (Fig. 3b). There is a positive correlation between the proportion of aneuploidies that are complex for each chromosome and their size (Fig. 3c), while several studies reported a negative correlation between chromosome size and occurrence of simple aneuploidy of whole chromosomes[37]. Additionally, we found an increasing proportion of complex aneuploidies with increasing ploidies (Fig. 3d), as was described

**Fig. 2 | Population structural diversity. a**, The outer donut plot indicates the number of SV of each type. The inner bar plot shows the repartition of SV among the 142 strains. **b**, Rarefaction curve showing the evolution of the number of nonredundant SV as a function of the number of sequenced strains. Inset plots show rarefaction curves per SV type. **c**, The numbers of SVs and SNVs/ indels are calculated relative to the reference genome (S288C). The categories 'heterozygous monosporic' and 'homozygous monosporic' correspond to monosporic isolates derived from sporulation of heterozygous and homozygous diploid strains, respectively. **d**, The allele frequency shows how SVs are shared among the strains. **e**, The values of 0 and 1 represent the relative positions of the centromeres and telomeres, respectively. **f**, The *x* axis has been truncated to 10 kb. The colors attributed to the different SV types are as in the other panels. **g**, The fold enrichments correspond to the ratio between the proportion of

breakpoints associated with a given ARS type and the proportion of the genome covered by the same ARS type. **h**, 'Intragenic' means that SVs are fully included within genes. 'Gene-containing' means SVs that contain at least one entire gene. 'Gene disruption' corresponds to SVs having one breakpoint located within a gene and the other breakpoint in an intergenic region. 'Gene fusion' indicates cases where the two SV breakpoints lie within two different coding sequences. In the essential column, n (no) and y (yes) mean not essential and essential genes, respectively. A manual review of 29 deleted genes, which are described as essential, revealed that they are in fact nonessential, are conditionally essential or are found deleted only at the heterozygous state. **i**, The numbers at the bottom indicate for each SV type the total number of SV–gene pairs and the number of pairs showing a substantial expression difference in the presence or absence of a given SV.

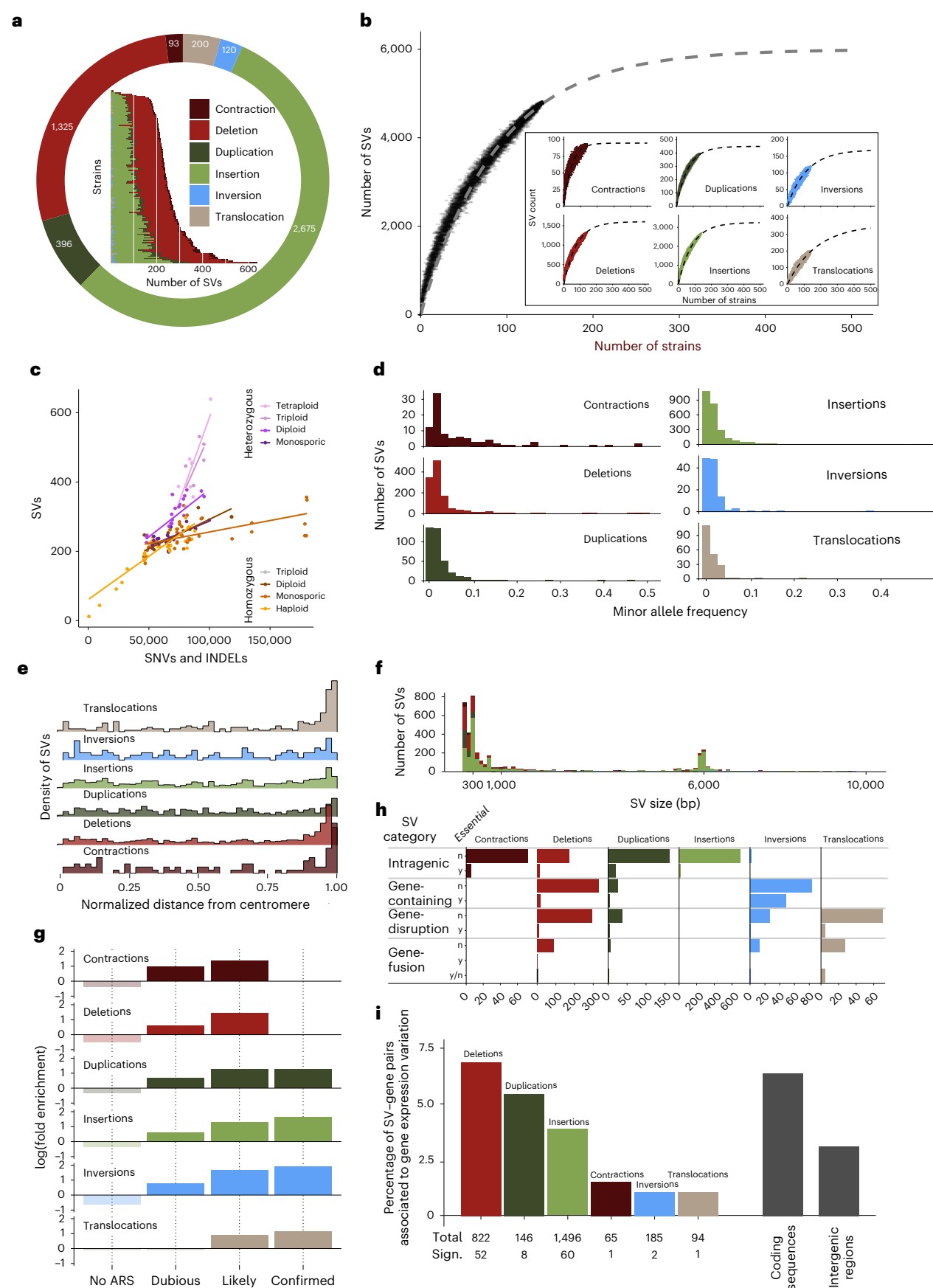

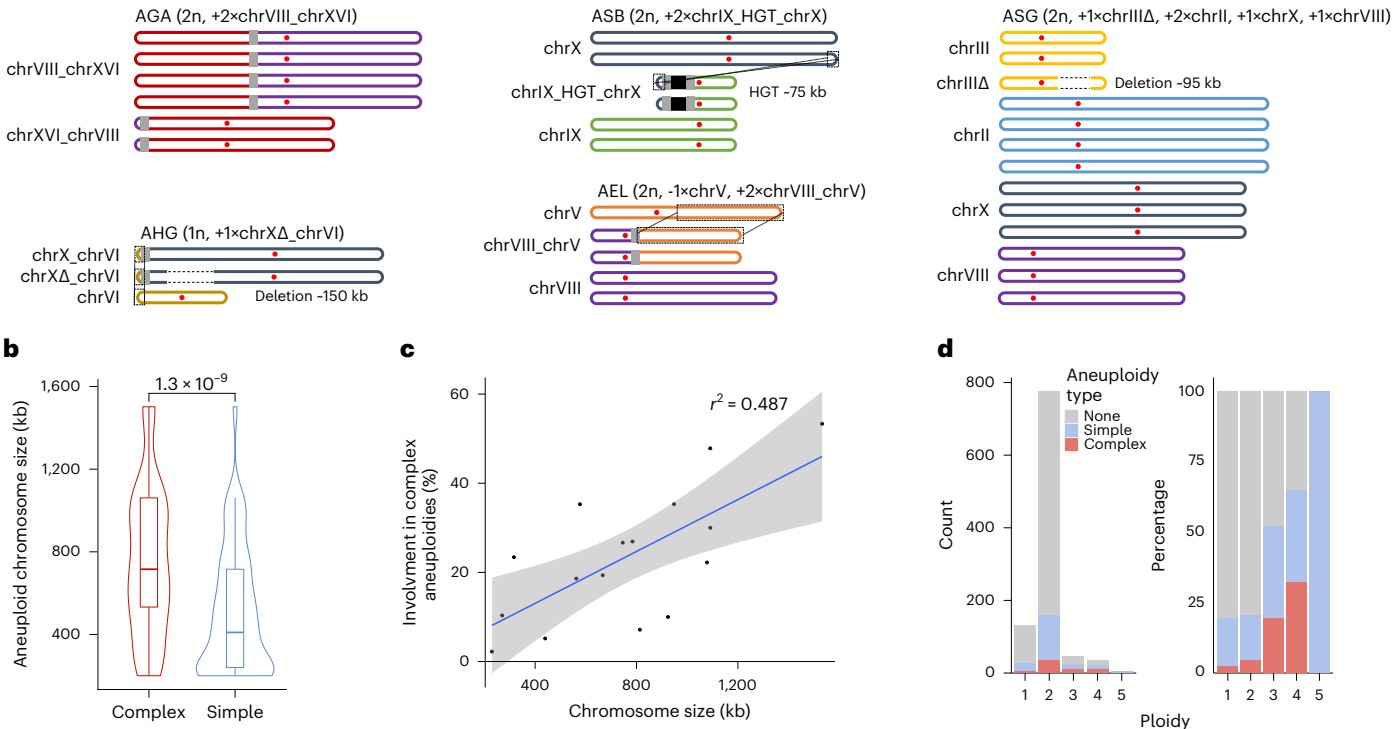

**Fig. 3 | Structure and prevalence of complex aneuploidies. a**, Schematics representing the chromosomal composition of complex aneuploidies. The red dots symbolize centromeres. The gray boxes indicate the translocation breakpoints. Dotted framings show duplicated regions, while dotted lines within chromosomes indicate deletions. The black box in ASB symbolizes the HGT region. **b**, Distribution of the sizes of chromosomes involved in complex ($n = 85$ different chromosomes) vs simple ($n = 379$ different chromosomes) aneuploidies. The horizontal lines in the boxplots correspond to the median, the lower and upper hinges correspond to the first and third quartiles and the whiskers extend up to 1.5 times the interquartile range. Two-sided Wilcoxon mean comparison *P* values are indicated. **c**, Percentage of the total chromosome aneuploidies that are complex as a function of their size. The coefficient of determination ($r^2$) was calculated with the lm method of the R program. The shaded error band represents the 95% confidence interval of the regression fit. **d**, Number and percentage of euploid, simple and complex aneuploid among 993 analyzed strains.

for simple aneuploidies[37]. These findings suggest that complex aneuploidies open an alternative adaptive route that would not be accessible to simple aneuploidies, by allowing increased copy number for genes located on large chromosomes.

**Evolutionary dynamics of complex genetic elements**
**Telomere length variation at TELO3L.** The mean telomere size varies by a factor of 4 across different isolates (Extended Data Fig. 4), from 166 bp in the CBS2183 wine strain (AFI) to 686 bp in the CLIB561 French dairy strain (BGN_3a). Some strains harbor homogeneous telomeres length across different chromosome extremities while others exhibit large variance. The average telomere length per strain correlates positively with variance (Extended Data Fig. 5a). We found no substantial correlation between telomere length and ploidy, heterozygosity or ecology (Extended Data Fig. 5b–d). We also examined telomere length variation between individual chromosome ends across the population of 100 strains. Despite a globally homogeneous distribution, TELO3L, and to a lesser extent TELO7R, are substantially longer than all other telomeres (Fig. 4a). The two same chromosome extremities were also described as the longest in W303 (ref. 38). The conservation of larger telomere size at TELO3L at the population level suggests that the underlying genetic determinants would be conserved because the species diverged from its last common ancestor at least ~180 KYA. We found that most ends carry a single copy of the core X element (67%, 3,036/4,528; Supplementary Table 11) and telomere length is substantially longer at chromosomal extremities devoid of this element,

but this difference is not visible at TELO3L (Extended Data Fig. 6a–c). We also found that telomere length was substantially larger when subtelomeres contained a *Ty5* element, but despite a specific enrichment, the TELO3L length is not influenced by the presence of *Ty5* (Extended Data Fig. 6d–f). Finally, we found that TELO3L is depleted in Y′ elements, being actually the poorest of all ends across the population (Extended Data Fig. 6g and Supplementary Note 3), and TELO3L that contain a Y′ element have substantially shorter telomeres than the ones without (Fig. 4b and Extended Data Fig. 6h,i) and this trend is unique to TELO3L among the 32 chromosome extremities (Extended Data Fig. 7). This finding suggests that the effect of the sequence that promotes the formation of longer telomeres at TELO3L would be specifically buffered by the presence of a Y′ element at this extremity.

**HGT regions constitute new telomeres.** Multiple HGT events have been reported in *S. cerevisiae*, but their mechanistic origin and accurate structure have remained elusive[31,39,40]. We characterized the structure and evolution of all large HGT regions known in *S. cerevisiae* (regions A–G; Supplementary Table 12). Although HGTs are enriched in domesticated strains, they are also present in wild isolates (Extended Data Fig. 8) showing that they can occur in natural environments and perhaps get amplified under anthropic conditions. An emerging feature shared by all HGT regions is that they are localized at telomeres, which implies that they must preserve or restore telomeric sequence and function upon their transfer (Supplementary Note 3). For instance, HGT region A (40 kb) has been transferred from an undefined *Torulaspora*

species[40] and is present at chromosome IX-L in DBVPG1608 (BLD_1a) and chromosome I-L in CBS422a (AAB; Fig. 4c and Extended Data Fig. 8). Additionally, the region A in AAB has a *Ty2* insertion, showing how these regions further evolved after the transfer. We inspected the telomeric repeats in the distal part of region A and observed that the more internal telomeric repeats of the *Torulaspora* donor species gradually shift toward the classic *S. cerevisiae* TG$_{1-3}$ repeats, with some intermediate repeats containing a mixed composition (Fig. 4c,d). This structure suggests that the *Torulaspora* repeats have seeded de novo telomere addition by telomerase to reconstitute a functional *S. cerevisiae*-like telomere.

**Whole-chromosome introgressions.** HP phasing of the MC9 (AIS) strain, isolated from Vino Cotto in Italy, revealed a hitherto unseen case of chromosome-scale introgression. One complete homolog of chrVI and a nearly complete homolog of chrVII introgressed from *Saccharomyces kudriavzevii* (Sk), exemplifying a unique hybrid karyotype (Fig. 4e). Recombination breakpoints occur in regions that have low divergence compared to the genome-wide average (1 SNPs every 4.78 bp, red dashed line, Fig. 4e). Overall, the peculiar AIS genome structure is hard to explain with current models of *Saccharomyces* genome evolution. The formation of a full Sc × Sk hybrid with a sequential loss of 14 Sk chromosomes and rediploidization of the corresponding Sc chromosomes or the partial transfer of two Sk chromosomes into an Sc strain represent two possible routes.

**Dynamics of tDNAs multigene families.** Multigene tDNA families are located in complex repetitive regions, as they serve as genomic targets for de novo transposition of the *Ty1* to *Ty4* elements[41], and therefore cannot be assembled by short-read genome sequencing. We identified 310 orthologous tDNA gene families that shared the same anticodon and were flanked by the same protein-coding genes, at least on one side (Supplementary Table 13). The tDNA repertoire is composed of 41 species of isoacceptors shared by all isolates. Two families underwent a mutation in the anticodon of one tDNA member (Supplementary Note 3). We found that 248 of 310 families were conserved in all 100 isolates, while the others were separated into two distinct categories depending on the number of strains they comprised. We observed 35 tDNA families in fewer than five strains, suggesting that they were acquired by recent tRNA gene gains, whereas 27 families were found in more than 90 strains suggesting recent losses in 1–10 isolates (Extended Data Fig. 9a), not necessarily shared by closely related strains as for instance the *tK(CUU)* that was lost ten times independently but never gained (Fig. 4f and Supplementary Table 14). In total, we found that 30 and 38 strains underwent 38 tDNA gene gains and 42 gene losses, respectively, and that all clades are affected by these events (Fig. 4f). Several strains accumulated multiple events (up to five in HN10 (BAM) isolated from rotten wood in China). Some clades preferentially accumulate one type of event, suggesting that functional constraints may favor the expansion or contraction of the tDNAs gene repertoire

(Fig. 4f). Interestingly, tDNAs that were recently gained are located closer to the chromosomal ends than the conserved or lost tDNAs (Extended Data Fig. 9b) with 17/35 newly gained tDNAs located in sub-telomeres while none of the 248 conserved genes are, suggesting that subtelomeres could serve as tRNA gene nursery where new copies are gained by segmental duplications associated with the junction of translocated segments from other chromosomes. The other 18 of 35 newly gained tDNAs that are located outside of the subtelomeric regions also result mainly from segmental duplications, either dispersed or in tandem.

**The genealogy of Ty elements.** We annotated all complete and truncated copies of retrotransposons and their solo-LTRs from the five families (*Ty1*–*Ty5*), as well as the *Tsu4* element, originating from a lineage related to *Saccharomyces uvarum* or *Saccharomyces eubayanus*[42] (Supplementary Table 11 and Supplementary Note 3). We observed that TEs are driving genome size variations, along with the Y′ elements (Extended Data Fig. 10a). The second largest genome (12.65 Mb) is from a monosporic isolate (AMM_1a) derived from a leaf tree in Taiwan (SJ5L12; Extended Data Fig. 10a) that underwent strong transposition activity[43] with a total of 120 complete and eight truncated elements while the median number is 14.5 (Extended Data Fig. 10b,c). The TE content is highly variable between isolates in terms of number and types of elements (Extended Data Fig. 10b,c), as previously described[43]. We identified 426 orthologous insertion sites shared between several genomes (that is, flanked by the same orthologous protein-coding genes, at least on one side, Supplementary Table 15). Their repartition across the population shows a U-shaped distribution with 50% being shared by less than 15 strains and 26% by more than 90 strains (Extended Data Fig. 10d). The most conserved sites are the most enriched in solo-LTR (Extended Data Fig. 10e and Fig. 4g), suggesting that inter-LTR recombination is common. The most conserved complete element is present in only 62 strains (Supplementary Table 15), and 118 insertion sites contain no full-length copy at all (Fig. 4g). The four closely related strains from the Malaysian clade (BMB, BMC_2a, UWOPS034614 and UWOPS052272) contain average numbers of solo-LTRs (about 390) and truncated Ty copies (between 6 and 9) but are completely devoid of full-length elements, suggesting that all functional copies were lost by recombination between LTRs. We confirmed that the Malaysian strains are among the most rearranged genomes, with 14 translocations and between 6 and 8 inversions per genome (Supplementary Table 6)[29,44], consistent with increased ectopic recombination between dispersed repeats. The frequent loss of complete elements by inter-LTR recombination is counterbalanced by an active process of de novo transposition. There are 61 sites only containing complete elements, two-thirds being found in a single isolate and the rest shared by a few strains (between 2 and 7) that are phylogenetic neighbors (Supplementary Table 15). This is particularly visible in clade 13 (lab-related strains) where 30 new insertions resulted from six recent independent insertion events (Extended Data Fig. 10f).

**Fig. 4 | Dynamics of complex regions. a**, Crossbars within violin plots indicate individual mean telomere lengths, and the horizontal line shows the global mean telomere length across all ends (*n* = 100 independent strains in each boxplot). No correction for multiple testing was applied, but the false discovery rates were estimated by calculating the proportion of false positives with 1, 2 and 3 stars black for the rank product and gray for the rank sum test) corresponding to *P* < 0.05, 0.01 and 0.001, respectively. **b**, Crossbars indicate TEL03L mean lengths (*n* = 100 independent strains in each boxplot). Two-sided Wilcoxon mean comparison *P* values are indicated. **c**, Region A was transferred from a *Torulaspora* species to BLD_1a and AAB strains and interrupted by a *Ty2* insertion in AAB. The most inner part of the telomeric repeats is close to the *Torulaspora* repeats (AAGGTTG**A/T**GGTGT[50]), while the distal portion consists of the *Saccharomyces* (TG$_{1-3}$). **d**, Telomere repeats gradually transition from *Torulaspora*-type to *S. cerevisiae*-type. The colors correspond to the repeat types

presented in **c**. **e**, *S. cerevisiae* and *S. kudriavzevii* chromosomes are represented in blue and in dark red, respectively. Both breakpoints on chromosomes VI and VII occur in regions that have low sequence divergence compared to the genome-wide average (red dashed line). **f**, The topology of the tree is the same as in Fig. 1. tRNA gene gains and losses are represented in dark blue and orange, respectively. Anticodon modifications are written in light blue. Strain names are colored in dark blue for gains, orange for losses and in mauve when different types of events co-occurred. **g**, Chromosomal repartition of all types of transposable elements across the 100 de novo assembled genomes (top). For isolates harboring several types of elements in a single region, the complete Ty is preferentially represented, followed by the truncated Ty. Chromosomal repartition of complete Ty elements (bottom). Only one element is plotted per isolate. For isolates with several families in a given insertion site, the family found in the reference genome was preferentially represented.

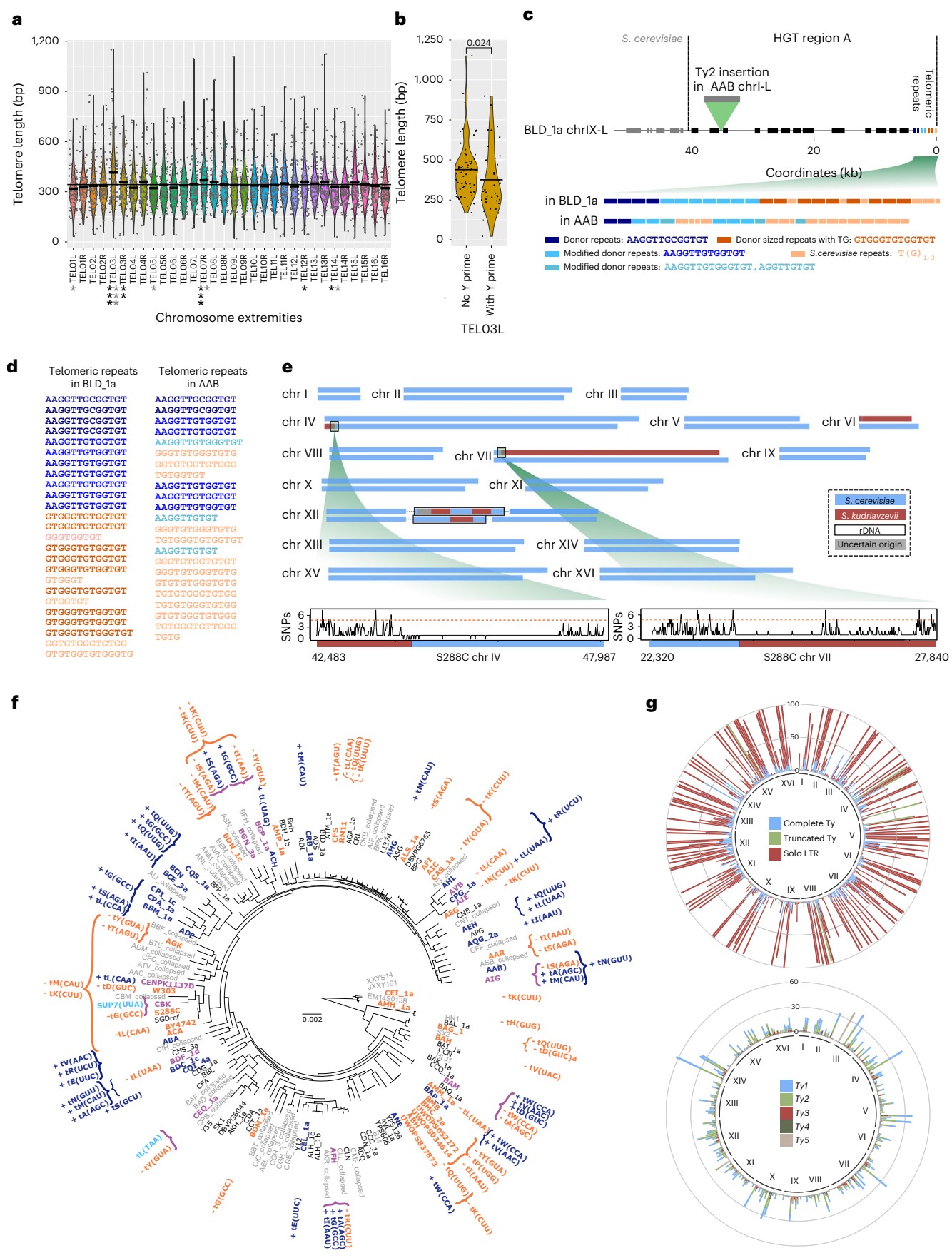

## Discussion

Using T2T assemblies of a large panel of *S. cerevisiae* strains, we captured a large fraction (80%) of the species structural diversity. We estimated that accessing the missing events would require an additional ~360 strains. We showed that SVs can impact the expression of genes located nearby. Additionally, we found that SVs have the potential to increase the gene repertoire diversity, calling for a pangenome paradigm shift that enables the functional characterization of accessory genes[45]. The true contribution of both SVs and accessory genes to the missing heritability remains to be quantified but the ScRAP represents a critical genomic resource toward this goal.

We found a median of 240 SVs (>50 bp) per genome, which represents an average density of 1 SV every 50 kb. By comparison, each human genome would contain >20,000 SVs[46], which corresponds to approximately 1 SV/150 kb, that is, three times lower than in *S. cerevisiae*. In other eukaryotes that benefit from pangenome data, the SV density scales from 1 SV/90 kb in *Drosophila*[47] (likely underestimated because only >100 bp euchromatic SVs were considered), 1 SV/38 kb in soybean[15], 1 SV/17 kb in rice[8] and up to 1 SV/4 kb in silkworm[17]. We also found a clear positive correlation between the numbers of SVs and SNVs/indels accumulating within genomes. It has been proposed that a genomic clock would coordinate the pace of fixation between amino acid substitutions and large-scale rearrangements in bacteria and yeast[48,49]. However, this clock seems to tick at a different pace depending on the ploidy and zygosity levels of the genome. SVs preferentially accumulate in heterozygous and higher ploidy genomes (Fig. 2c). One possibility would be that SVs are better tolerated in higher ploidy genomes as their deleterious effects (for example, gene deletion and dosage imbalance) are more efficiently buffered. Alternatively, the rate of SV formation might increase with ploidy, as was suggested for aneuploidies[37].

In the near future, high-quality de novo assemblies of thousands of individuals will generate a unified, complete and accurate representation of the species genomic diversity. Beyond the analysis reported here, the ScRAP provides a solid foundation for this purpose and will drive the transition to a pangenome free from reference bias.

## Online content

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

## Methods

### ScRAP strain selection

The full panel is composed of three distinct datasets as follows: (1) 100 newly sequenced and de novo assembled genomes, (2) 18 re-assembled genomes using previously available raw Nanopore read data[25] and (3) 24 publically available complete genome assemblies, including the S288C reference genome[22–24,26,28,29,51,52] (Supplementary Fig. 1). The rationale for the selection of the 100 strains de novo sequenced in this study was based mainly on the knowledge gained from the 1,011 genome project[31,53]. We selected one per clade and subclades, with a good sporulator phenotype. We selected some strains with a known signature of SV (for example, AIF with segmental duplications). The AIS strain that contained chromosome-scale introgression was first detected in the 1,011 work but excluded because of its complex genome structure. We also selected strains known to carry large HGT events. The 31 diploids (ten nearly homozygous and 21 highly heterozygous) that were unable to sporulate or produce viable spores were sequenced in their original ploidies. Note that, as an exception, BAF was sequenced as a diploid despite the fact that it is sporulating well and has good spore viability.

### DNA extraction and sequencing

**ONT library preparation and sequencing.** We grow yeast cells in 10–15 ml yeast peptone dextrose (YPD) media at 30 °C overnight (220 rpm). A total number of cells less than $7 \times 10^9$ were used for DNA extraction. High molecular weight (HMW) DNA was extracted by QIAGEN Genomic-tip 100/G according to the 'QIAGEN Genomic DNA handbook' for Yeast. DNA quantity and length were controlled by Qubit dsDNA HS Assay and pulsed-field gel electrophoresis (PFGE), respectively. Library preparation and ONT sequencing were performed based on the protocol of '1D Native barcoding genomic DNA with EXP-NBD104 and SQK-LSK108' when using FLO-MIN106 MinION flow cells and protocol of '1D Genomic DNA by Ligation with EXP-NBD104 and SQK-LSK109−PromethION' when using the V2 FLO-PRO002 flow cell. These protocols are available from Oxford Nanopore Technologies Community.

For sequencing library preparation, up to 2 µg of HMW DNA per sample was used to start library preparation. DNA repair and end preparation were performed using the NEBNext FFPE DNA Repair Mix with the following reaction setup: 48 µl DNA, 3.5 µl NEBNext FFPE DNA Repair Buffer, 2 µl NEBNext FFPE DNA Repair Mix, 3.5 µl Ultra II End Prep Reaction Buffer and 3 µl Ultra II End Prep Enzyme Mix; 20 °C for 15 min followed by 65 °C for 15 min. Afterward, the DNA size selection was carried out using AMPure XP Beads (1:1 ratio) followed by native barcode ligation (22.5 µl DNA, 2.5 µl native barcode provided by EXP-NBD104 kit and 25 µl Blunt/TA Ligase Master Mix; 25 °C for 20 min). After another round of AMPure XP bead clean-up (1:1 ratio), the samples were pooled together and adaptors were ligated for the pooled sample at 25 °C for 15 min (65 µl DNA, 5 µl adapter mix II (AMII) provided by EXP-NBD 104 kit, 20 µl NEBNext Quick Ligation Reaction Buffer and 10 µl Quick T4 DNA Ligase; 25 °C for 15 min). The adaptor-ligated DNA was cleaned up by adding a 0.4× volume of AMPure XP beads followed by incubation for 5 min at room temperature. When using the SQK-LSK108 kit for FLO-MIN106 MinION flow cells, twice 140-µl adapter bead binding buffer (ABB) washes were performed. When using the SQK-LSK109 kit for FLO-PRO002 flow cells, twice 250-µl L fragment buffer washes were performed. The final library was eluted in 15 µl Elution Buffer and loaded into the MinION or PromethION flow cells according to the ONT manuals. Raw fast5 files were basecalled using Guppy (version: 3.4.5) followed by Porechop (version: 0.2.4;github.com/rrwick/Porechop) removal of adapters and barcodes. The entire project generated close to 204 Gbp of Nanopore sequencing data. Sequencing statistics are detailed in Supplementary Table 16. For fast5 file storage/sharing, single-fast5 files were stripped of basecalling data using Picopore (version: 1.2.0; github.com/scottgigante/picopore), ensuring all files contain only data necessary for re-basecalling. Next, single-fast5 files were converted into multi-fast5 files using the ont-fast5-api (version: 0.3.2; github.com/nanoporetech/ont_fast5_api) single_to_multi command, followed by the fast5_subset command to generate strain-specific fast5 files containing fast5 files for all reads within each strain-specific fastq file. This was done to reduce the complexity of reanalysis using fast5 files from strains run with both/either multiple barcodes and across several flow cells and remove fast5 files for reads of insufficient quality. All adapter/barcode-free fastq files and their associated strain-specific fast5 files are available under the accession PRJEB50706/ERP135326.

**Illumina sequencing.** We grew yeast cell cultures overnight at 30 °C in 20 ml of YPD medium until the early stationary phase. We collected cells by centrifugation and extracted total genomic DNA using the QIAGEN Genomic-tip 100/G according to the manufacturer's instructions. Genomic Illumina sequencing libraries were prepared with a mean insert size of 280 bp and subjected to paired-end sequencing (2 × 100 bp) on Illumina HiSeq 2500 sequencers. All paired-end Illumina reads are available under the accession PRJEB50706/ERP135326.

### Pipelines for genome assembly, HP phasing, genome annotation and SV detection

All pipelines are detailed in the Supplementary Methods.

### Analysis of complex regions

**Subtelomere dynamics.** The subtelomeric regions were annotated and named in the same way as our previous study proposed[29]. Manual examination and adjustment were further applied to curate subtelomeric ends with incomplete sequence information or substantial reshuffling (Supplementary Table 17).

**tDNAs and Ty elements.** We defined a consistent set of 100 haploid or homozygous genome assemblies for the analyses of the dynamics of tDNAs and Ty elements by first excluding the diploid, triploid and tetraploid phased assemblies because they contained numbers of annotated tDNA and Ty copies that were proportional to their ploidy and, therefore, difficult to compare with haploid and collapsed genome assemblies (Supplementary Table 11). We also removed eight haploid genomes from one specific study[22] because they contained a much lower number of tDNAs than all other genomes of the dataset, probably indicative of local assembly errors. We finally excluded collapsed assemblies from heterozygous genomes because they showed some discrepancies with their cognate phased assemblies, suggesting possible assembly problems in these complex regions.

### Telomere detection

We developed Telofinder[54] (https://telofinder.readthedocs.io/en/latest/) to determine the chromosomal location and size of telomere sequences in yeast genomes assemblies. Telomere detection is based on the calculation of both the DNA sequence entropy and the proportions of the 'CC,' 'CA' and 'AC' dinucleotides in a 20-bp sliding window. Telofinder outputs two csv and two bed files containing the telomere calls and their coordinates, either as raw output or after merging consecutive calls. We ran Telofinder (version: 1.0; options: -s -1) on all 394 de novo and 24 previously available nuclear genome assemblies to scan entire genome sequences.

### Aneuploidy detection

**Aneuploidy detection in the ScRAP.** Illumina reads were aligned to the reference genome using BWA-MEM (version: 0.7.17), and coverage was calculated using BEDTools genomecov (version: 2.27.1; options: -d -ibam). Genome coverage was then visualized for each strain separately with centromere positions. Aneuploidies were then manually annotated. To validate complex aneuploidy structures, we used additionally both nanopore reads and genome assemblies. Nanopore reads were aligned to the reference using minimap2 (version: 2.17)

and visualized using tablet[55]. Raw and finalized genome assemblies were aligned with MUMmer nucmer (version: 4.0.0beta2) against the S288C reference assembly and other assemblies and visualized as a dotplot. Additionally, by analyzing raw genome assemblies, four complex aneuploidy chromosomes that were completely or partially assembled, at least containing the complex region, were identified and extracted (CBS1586/AHG +1×chr10c; CBS457/AIF +1×chr11c; CBS4255/ASB +2×chr9c; CBS1489/ASG +1×chr3c).

**Aneuploidy detection in a dataset of 909 strains.** The aneuploidy detection pipeline[56] using Illumina data from refs. 31,36 is available at https://github.com/SAMtoBAM/aneuploidy_detection and consists of the following steps:

1. Illumina reads were aligned with BWA-MEM (version: 0.7.17), and the coverage was calculated using BEDTools genomecov (version: 2.27.1; options: -d -ibam).

2. Coverage was binned for calculating the median coverage in 30-kb bins with a 10-kb stepsize using both BEDTools makewindows and map, during which regions covering 15 kb (half a window) from each chromosome end were removed to reduce telomere mapping/variation issues.

3. Coverage was normalized by the genome-wide median, and candidate regions were extracted if they deviated by ±0.7*(1/$n$). 0.7 gives some leniency to coverage deviation considered sufficient for a change in copy number.

4. Deviating bins were aggregated, allowing for a gap of <=10 kb (the size of 1 slide).

5. Aggregated bins were split into two types depending on whether they overlapped a centromere or not, called centromere related (CR) and noncentromere related (NCR), respectively.

6. The size of CR regions was increased to the sum of all regions within the same chromosome with the same deviation, for example, +1, +2, −1, etc., to give a CR-sum. We then removed any CR-sum <50 kb and calculated the difference between this CR-sum and the chromosome size (minus the 30 kb removed from the ends).

7. All CR-sums with a difference in size >100 kb (that is, a the CR does not cover region/s summing to 100 kb or more) were labeled as complex and the rest as simple

8. Plot the normalized coverage of all aneuploidies and manually curate the list to remove false positives and adjust the complex-simple classification.
   During the manual curation, 34 complex stayed complex, 347 simple remained simple, four complex were removed as false-positive aneuploidies (due to the impact of the 'smiley-effect'), 32 moved from complex to simple and ten moved from simple to complex. The move from simple to complex was always due to the threshold of size (the size difference of these ten CRs was 96, 90, 82, 80, 72, 72, 66, 66, 50 and 40 kb). All examples of this reclassification from simple to complex were distinct.

9. Identify any NCRs >100 kb that are present within a strain containing an aneuploidy detected above. Label as complex aneuploidy-related and use in the less conservative estimate of complex aneuploidy count.

The newly generated large aneuploidy dataset overlaps, by strain and chromosome, 88% (303/343) with that of the ref. 31 dataset. This leaves only 40 aneuploidies (12%) not redetected. Of these 40, manual inspection identified that nine were clearly false positives in ref. 31, eight came from the same strain and are an issue of defining whether eight chromosomes were lost or the other eight were gained, one contains a large increase in coverage close to the centromere but not covering and three contain a slight change in coverage but are well below the set threshold for indicating a change in copy number. Therefore, only

19 missed aneuploidies are likely true false negatives in the new dataset. The overlap between datasets additionally shows 120 previously undetected aneuploidies in the new dataset. Of these 120, 35/120 (29%) are complex aneuploidies, as compared to 9/303 (2%) found within the overlap.

## SVs and gene expression impact

**Determination of SV–gene pairs.** SV–gene pairs were evaluated using BEDTools (version: 2.27.1). For SVs overlapping CDS, BEDTools intersect was used to identify the pairs. A supplementary awk filter was applied to specifically identify CDS fully located within SVs. For SVs within intergenic regions, BEDTools closest was used to identify the pairs, either by identifying the two SVs closest to a CDS in the case of indels (using the −io and −id or −iu options) or by identifying the CDS closest to each boundary of an SV, as well as the ones that might overlap SV boundaries. In the case of inversion events, only the pairs involving the genes closest or overlapping the associated inversion breakpoints were investigated. Additionally, in the case of indels, SVs were associated with a CDS only if they were located in the intergenic space between the observed CDS and the next, both upstream and downstream.

**Evaluation of the impact of SVs on the expression of neighboring genes.** For each of the SV–gene pairs obtained at the previous step, the 51 studied strains were split into the following two groups: strains with and without the SV. Then, the expression values of the studied gene in each of the strains were ranked and normalized between 0 and 1 and then used to evaluate the differential expression by performing a two-sided Wilcoxon–Mann–Whitney test between the with- and without-SV groups. This statistical analysis was performed using R (version: 3.5.1).

## Phylogenetic reconstruction

**Ortholog-based nuclear tree.** For the nuclear genome, the proteome sequences of 181 input genomes (with 23 outgroup *Saccharomyces* species) were used for phylogenetic analysis. A total of 1,612 one-to-one nuclear ortholog groups were identified by Proteinortho (version: 6.0.25; options: --check -selfblast -singles). For each ortholog group, the protein and CDS alignment were generated by MACSE (version: 2.04; options: -prog alignSequences -gc_def 1 -seq $i.species_relabeled.fa -out_NT $i.macse_NT.aln.fa -out_AA $i.macse_AA.aln.fa and -prog exportAlignment -align $i.macse_NT.aln.fa -codonForFinalStop --- -codonForInternalStop NNN -codonForInternalFS NNN -codonForExternalFS --- -charForRemainingFS - -out_NT $i.macse_NT.aln.tidy.fa -out_AA $i.macse_AA.aln.tidy.fa). A concatenated supermatrix of the 1,612 ortholog-based CDS alignment was further generated with different partitions defined corresponding to different ortholog groups. This supermatrix and its associated partition definition were used for maximal likelihood tree building by IQtree (version: 1.6.12; options: -spp $prefix.concatenated.cds.partition.txt -s $prefix.concatenated.cds.tidy.fa -m MFP -bb 1000 -alrt 1000 -nt $threads -pre $prefix.iqtree -safe). In total, 1,000 rounds of ultrafast bootstrap (UB) and approximate likelihood-ratio test (aLRT) were used to assess the branch supports.

**SNP-based nuclear tree.** The input VCF file (matrixSam.snp.vcf.gz) was generated using GATK4's HaplotypeCaller and GenotypeGVCFs (version: 4.1.8.1) with BWA-MEM (version: 0.7.17) aligned Illumina reads. The resulting multisample VCF was then filtered for variants with a Quality-by-Depth, StrandOddsRatio, FisherStrand, Mapping Quality, MappingQualityRankSum and/or ReadPosRankSum of more than two s.d. from the average. Finally, variants were removed from regions labeled as repetitive by RepeatMasker and/or in ref. 57 to generate the final VCF.

We used the vcf2phylip[58] python script (https://github.com/edgardomortiz/vcf2phylip; versions: 2.8; options: -I $input_vcf

−resolve-IUPAC -o S288C −fasta −output-prefix) to convert the VCF file into the fasta format. The corresponding SGDref entry in fasta format was extracted based on the reference allele column of the input vcf file. MAFFT (version: 7.471; options: --auto --thread $threads --preservecase --addfragments) was used to align these two fasta files by using the extracted SGDref entry as the reference sequence for alignment. The resulting alignment was further filtered by ClipKIT (version: GitHub commit cccc8bf; options: -m gappy). The filtered alignment was fed into IQtree for tree building (version: 1.6.12; options: -s $prefix.fasta -m GTR+ASC -bb 1000 -alrt 1000 -nt $threads -pre $prefix.iqtree -safe). A thousand rounds of UB and aLRT were used to assess the branch supports.

**SV-based nuclear tree.** The input SV VCF file (homo_and_hetero_noDoublonsInCoordinates.vcf.gz) was generated using the nonredundant SV dataset. Based on the presence/absence information of these identified SVs in each assembly entry, a phylip-formatted 0/1 matrix was generated accordingly and used for tree building. IQtree (version: 1.6.12; options: -s $prefix.phylip -st MORPH -m MK+ASC -bb 1000 -alrt 1000 -nt $threads -pre $prefix.iqtree -safe) was used to generate the phylogenetic tree. A thousand rounds of UB and aLRT were used to assess the branch supports.

**Phylogenetic tree processing and visualization.** For the phylogenetic trees generated above, tree-based operations such as rerooting, branch trimming and tip label extraction were performed by the nw_reroot, nw_prune, nw_labels tools from the Newick Utilities package (version: 1.6.0). Tree visualization was performed by the R package ggtree (version: 3.2.1). Cophylo comparison was conducted by the R package phytools (version: 1.0-3). The distance between trees was evaluated in terms of the quantity of information that the trees' splits hold in common with the clustering information distance implemented in the R package TreeDist (version: 2.4.1).

### Molecular dating
We used the formula for molecular dating published earlier[59]. We considered 100 and 365 generations per year to bind our estimations, as previously suggested[60]. The value of the mutation rate of 2.31072123540072E-10 was calculated as the average of the rates for homozygous and heterozygous lines reported previously[61]. The pairwise distances between strains were calculated using MEGA11 (version: 11.0)[62] as p-distance, using only the fourfold degenerate sites. To determine if a codon position is a fourfold degenerate site, we scan over every codon and codon position (that is, first, second and third positions) based on NCBI's codon table (https://www.ncbi.nlm.nih.gov/Taxonomy/Utils/wprintgc.cgi) based on the CDS alignment of each orthologous gene group. All codon positions corresponding to fourfold degenerate sites were concatenated together to form the fourfold degenerate site alignment of the corresponding CDS alignment. The fourfold degenerate site alignment of all 1-to-1 ortholog CDSs was further concatenated to form a super alignment of fourfold degenerate sites.

### Reporting summary
Further information on research design is available in the Nature Portfolio Reporting Summary linked to this article.

### Data availability
All sequencing data and assembly/annotation files were deposited in the European Nucleotide Archive (https://www.ebi.ac.uk/ena/browser/home) under the umbrella project PRJEB59869. The project accession for the raw sequencing data (fast5, nanopore fastq, Illumina fastq) is PRJEB50706. The assembly/annotation accessions are PRJEB59413, PRJEB59129, PRJEB59231, PRJEB59232 and PRJEB59230 for unphased nuclear, HP1, HP2, HP (for polyploids) and mitochondrial

assemblies, respectively. Each accession for individual assemblies is indicated in Supplementary Table 1 (nuclear) and Supplementary Table 3 (mitochondrial).

### Code availability
All published and/or publicly available software used in the study, with their version numbers and their reference for downloading, are noted in the methods and Supplementary Methods sections as well as in the Reporting Summary. Custom scripts developed in this study are Telofinder (https://doi.org/10.5281/zenodo.8063924)[54] that is also available at https://github.com/GillesFischerSorbonne/telofinder, the aneuploidy detection pipeline (https://doi.org/10.5281/zenodo.8068318)[56] that is also available at https://github.com/SAMtoBAM/aneuploidy_detection, the HP phasing pipeline (https://doi.org/10.5281/zenodo.8068328)[63] that is also available at https://github.com/SAMtoBAM/PhasedDiploidGenomeAssemblyPipeline and the script for generating the nonredundant SV dataset (https://doi.org/10.5281/zenodo.8068284)[64] that is also available at https://github.com/SAMtoBAM/MUMandCo/tree/master/nonredundant_population_datasets.

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

### Acknowledgements
We thank B. Dujon and B. Llorente for their valuable feedback on the paper. This work was supported by the Agence Nationale de la Recherche ANR-16-CE12-0019 (to G.L., J.S. and G.F.) and ANR-18-CE12-0004 (to G.L. and G.F.). This work was also partially supported by the ANR-15-IDEX-01 (to G.L.), Fondation pour la Recherche Médicale (EQU202003010413 to G.L.), the European

Research Council (ERC Consolidator Grant 772505 to J.S.), the Guangdong Basic and Applied Basic Research Foundation (2019A1515110762 to J.-X.Y.), Guangdong Pearl River Talents Program (2019QN01Y183 to J.-X.Y.), and National Natural Science Foundation of China (32070592 to J.-X.Y.). J.S. is a Fellow of the University of Strasbourg Institute for Advanced Study (USIAS) and a member of the Institut Universitaire de France.

## Author contributions

J.S., G.L. and G.F. conceived the study. S.O.D., N.A., C.C., S.D., T.F. and J.L. performed experiments. S.O.D., J.-X.Y., O.A.S., T.C., M.D.C., F.D., T.F., A.F., E.K., J.L., Z.M., L.T., J.S., G.L. and G.F. analyzed results. S.O.D., J.-X.Y., J.S., G.L. and G.F. wrote the paper. All authors reviewed and contributed to the final version of the paper.

## Competing interests

The authors declare no competing interests.

## Additional information

**Extended data** is available for this paper at https://doi.org/10.1038/s41588-023-01459-y.

**Correspondence and requests for materials** should be addressed to Joseph Schacherer, Gianni Liti or Gilles Fischer.

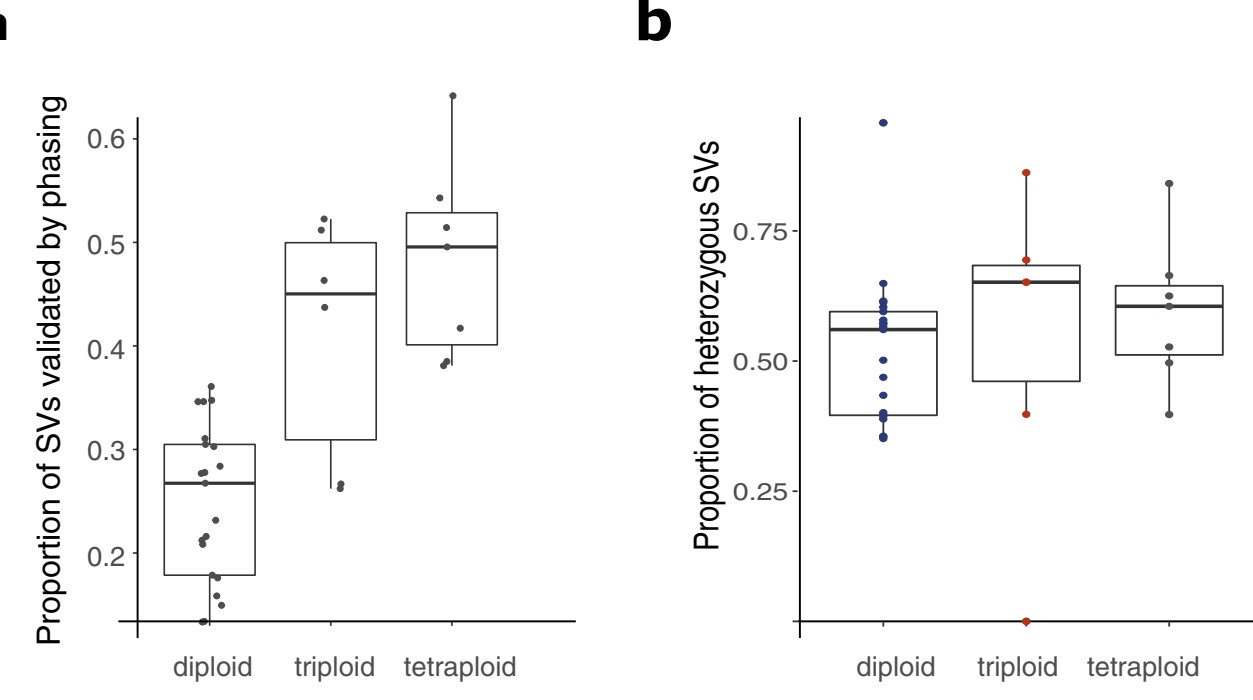

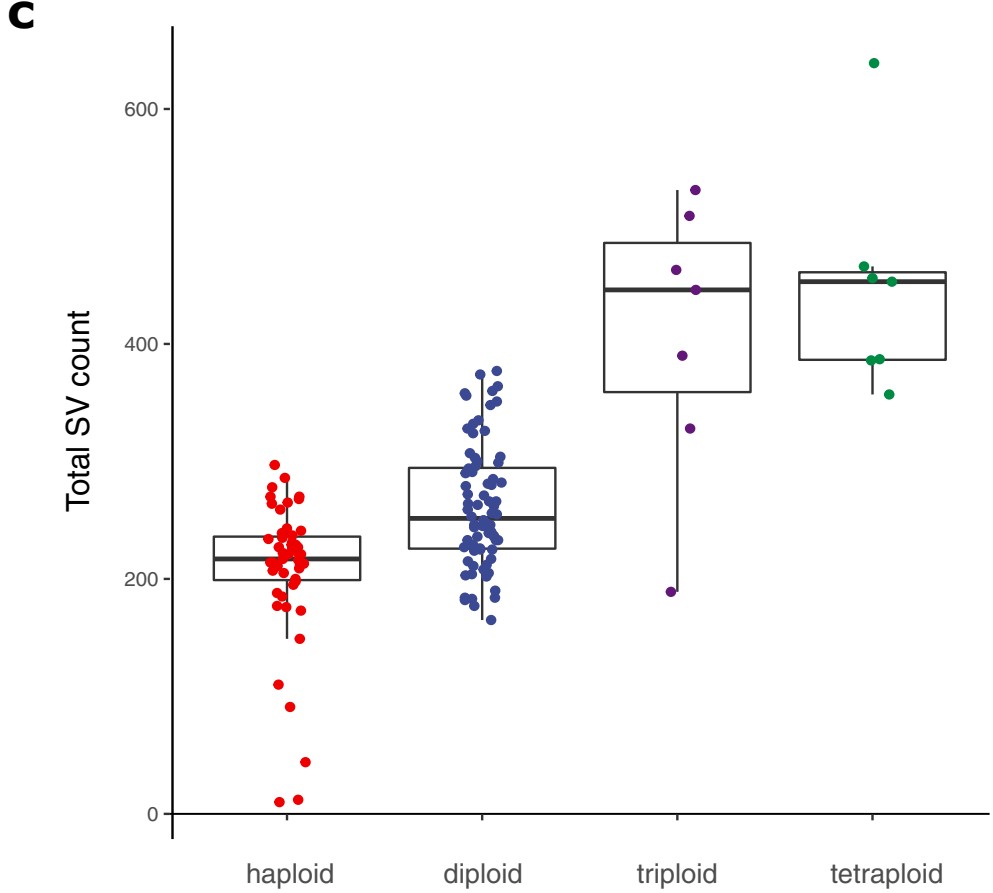

**Extended Data Fig. 1 | See next page for caption.**

**Extended Data Fig. 1 | Impact of zygosity and ploidy on SV detection.**
**a**. Impact of haplotype phasing on SV validation for the different levels of ploidy (n = 21, n = 6 and n = 7 for diploid, triploid and tetraploid strains, respectively).
**b**. Distribution of the number of heterozygous SV per strain split by ploidy. For diploids this was simply done by considering any variant that did not contain both HP1 and HP2 genomes as heterozygous (n = 21 strains). For polyploids, first the phased genomes were aligned to the reference, coverage was calculated around the region of the event and this coverage was then used to estimate the maximum number of haplotypes present. If the number of phased blocks validating the variant was fewer than the max haplotypes, the event was considered heterozygous (n = 6 and n = 7 triploid and tetraploid strains, respectively). The horizontal lines in the boxplots correspond to the median, the lower and upper hinges correspond to the first and third quartiles and the whiskers extend up to 1.5 times the interquartile range. **c**. Number of validated SV per strain split by ploidy (n = 51, n = 76, n = 6 and n = 7 for haploid, diploid, triploid and tetraploid strains, respectively). The horizontal lines in the boxplots correspond to the median, the lower and upper hinges correspond to the first and third quartiles and the whiskers extend up to 1.5 times the inter quartile range.

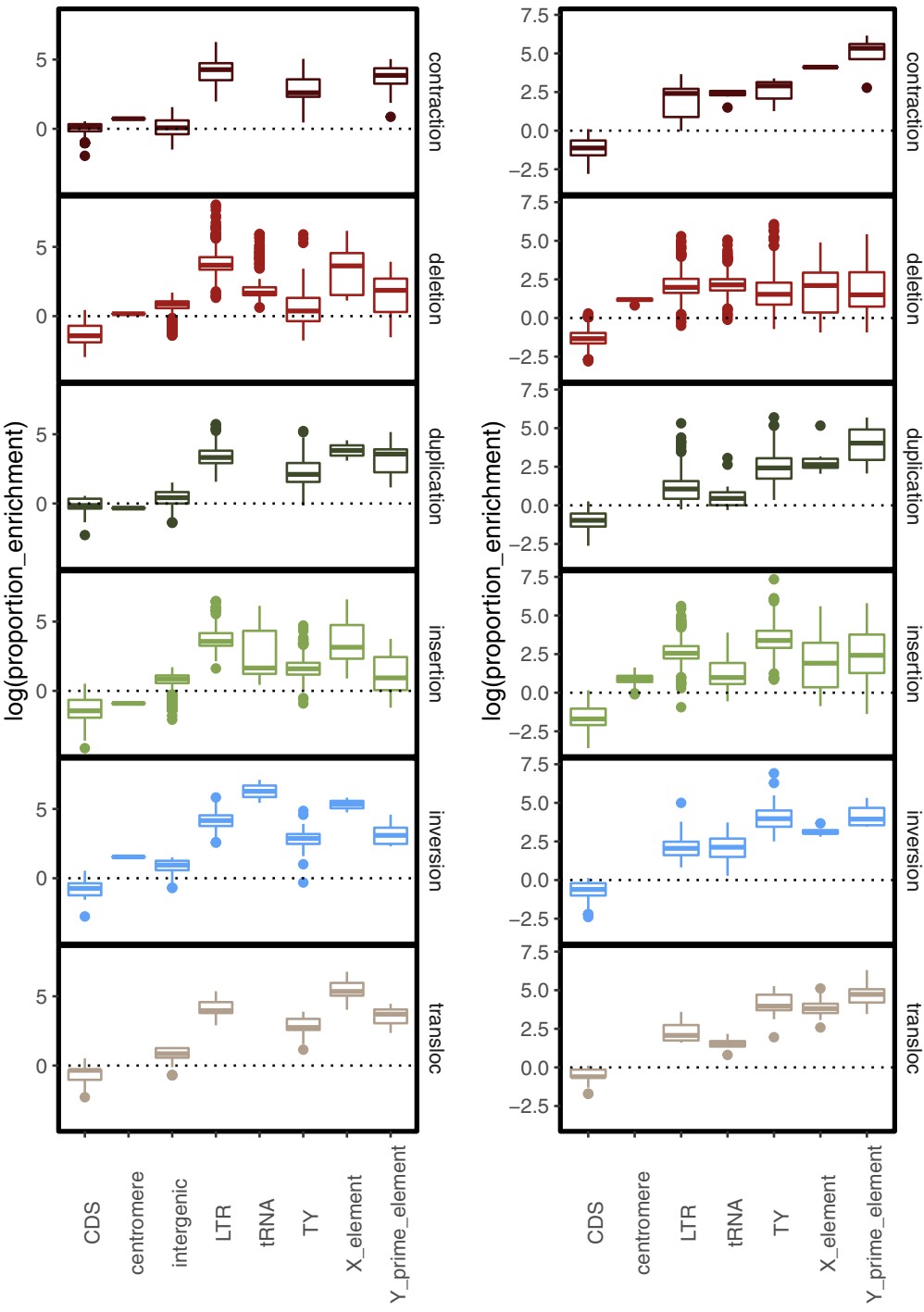

Genomic feature with breakpoint

Closest feature to intergenic breakpoint

**Extended Data Fig. 2 | Association between various genetic elements and SV breakpoints split by SV type.** The breakpoints of all SVs within every 'best' genome were matched to the corresponding gff annotation file. These were then compared to the genome wide proportion of events to calculate a per genome enrichment. This was done by getting the breakpoint-associated features (left: n = 1776 CDS, n = 5 centromere, n = 1743 LTR, n = 274 tRNA, n = 760 TY, n = 354

X_element and n = 498 Y_prime_element) and the closest element for intergenic breakpoints (right: n = 2041 CDS, n = 145 centromere, n = 1784 LTR, n = 1291 tRNA, n = 748 TY, n = 427 X_element and n = 278 Y_prime_element). The horizontal lines in the boxplots correspond to the median, the lower and upper hinges correspond to the first and third quartiles and the whiskers extend up to 1.5 times the interquartile range.

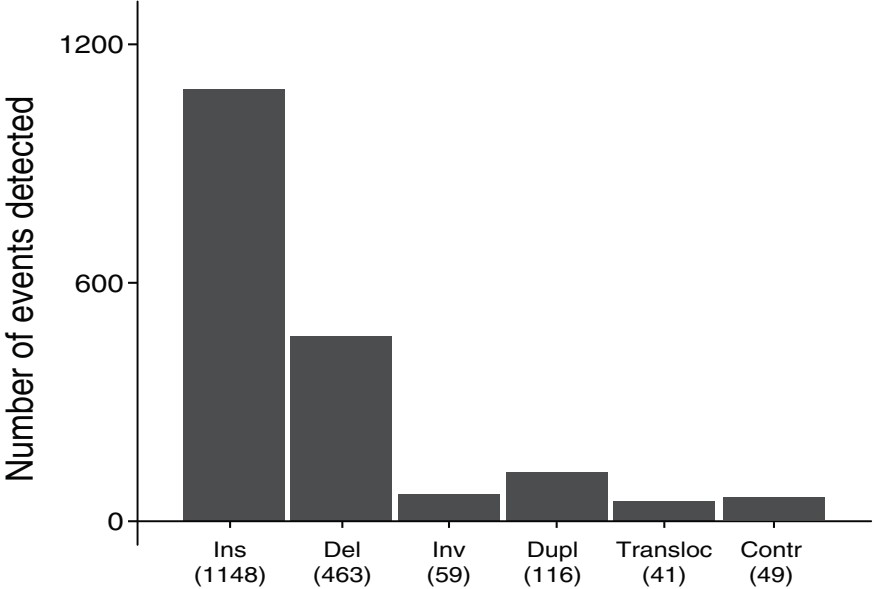

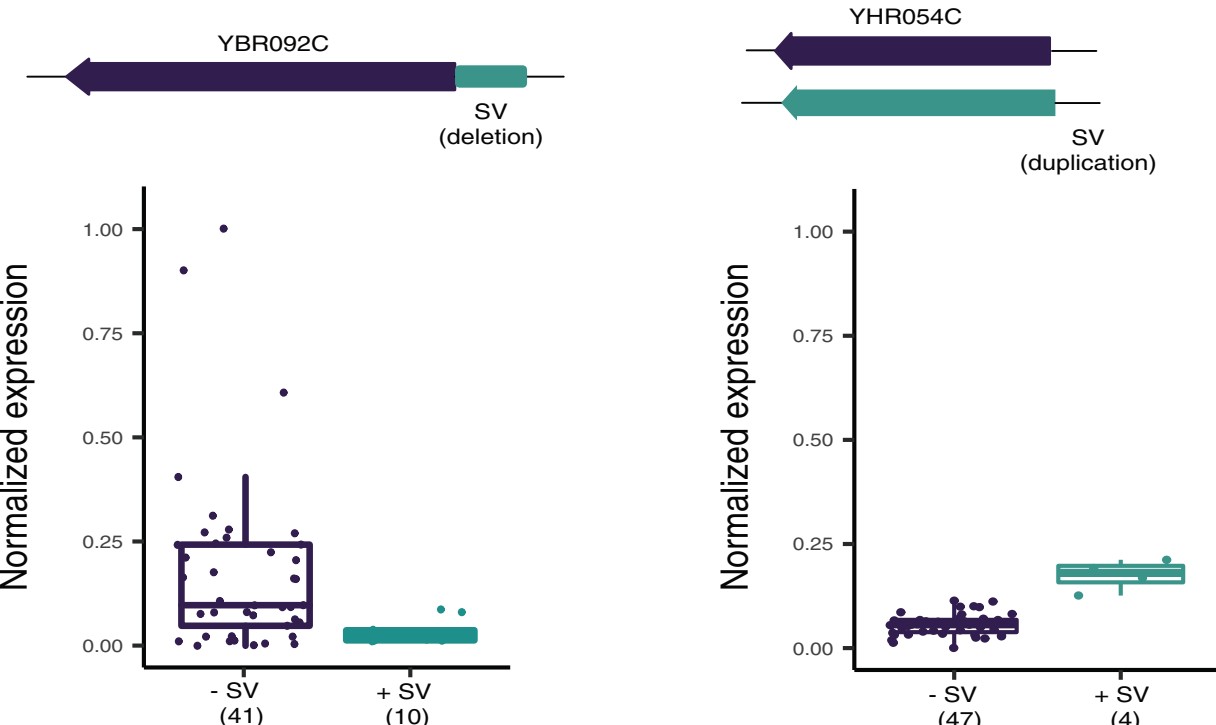

**Extended Data Fig. 3 | Relationship between SV and gene expression.**
**a**. Number of the different types of SVs present in the set of 51 isolates used to study the relationship between SVs and gene expression variation (Ins for insertions, Del for deletions, Inv for inversions, Dup for duplications, Transloc for translocations, and Contr for contractions). **b**. Comparison of the expression level allowed to test for the impact of the SV. Left panel, comparison of the presence (+SV) or absence (−SV) of a deletion event in the regulatory region of the ORF YHR043C. Right panel, comparison of the presence (+SV) or absence (−SV) of a duplication of the ORF YHR054C. The horizontal lines in the boxplots correspond to the median, the lower and upper hinges correspond to the first and third quartiles and the whiskers extend up to 1.5 times the interquartile range.

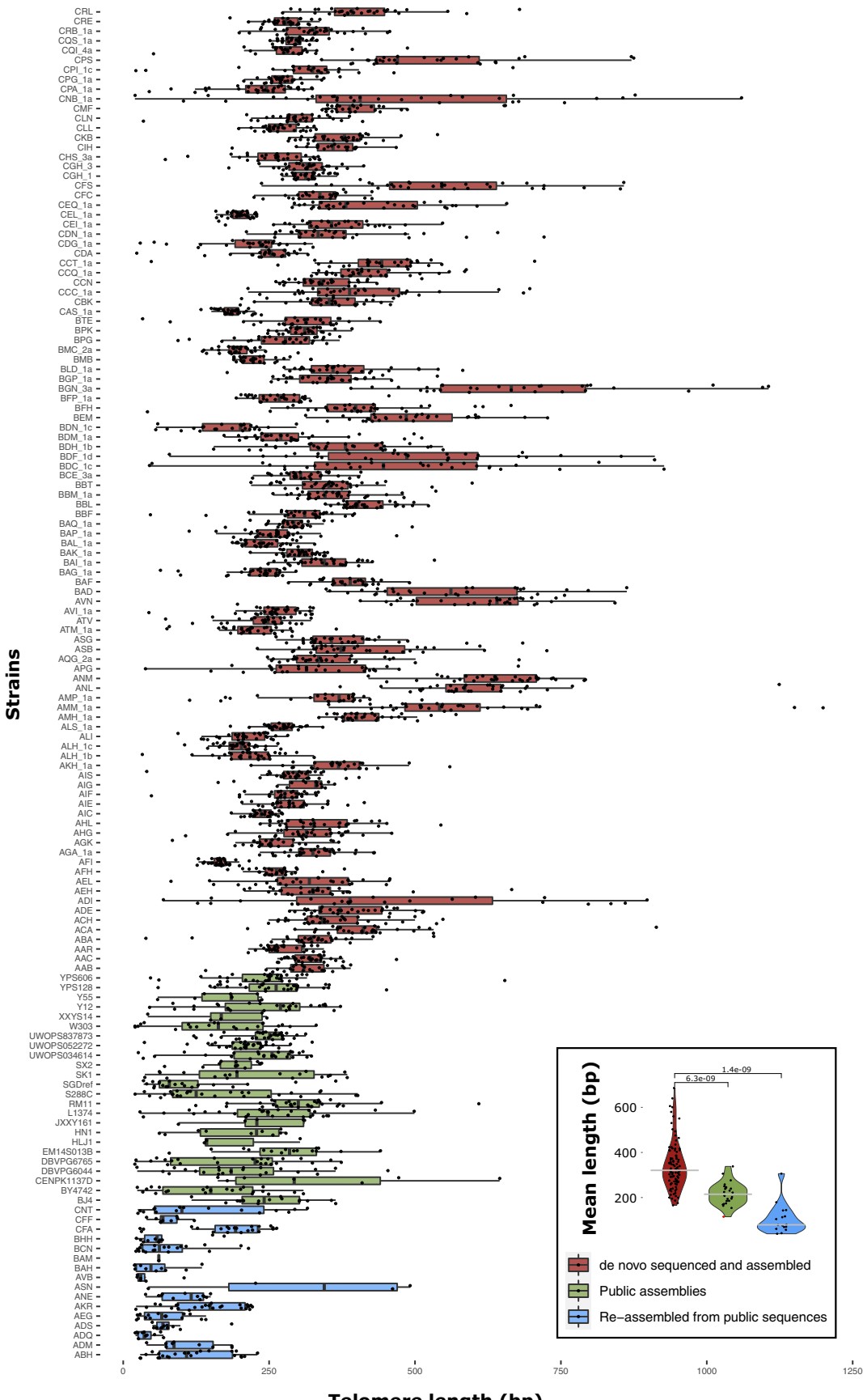

**Extended Data Fig. 4 | See next page for caption.**

**Extended Data Fig. 4 | Distribution of telomere length at all chromosome ends in each strain of the ScRAP split by dataset.** The vertical lines in the boxplots correspond to the median, the lower and upper hinges correspond to the first and third quartiles and the whiskers extend up to 1.5 times the interquartile range. For each of the 142 strains, the number of telomeres used to derive the boxplot is indicated in Supplementary Table 1. The inset plot shows the distribution of the average telomere length per strain for each dataset. Median values are indicated by the gray crossbars in each violin plot (for *de novo* assemblies n = 100 strains, for public assemblies n = 24 strains and for re-assemblies n = 18). Two-sided Wilcoxon mean comparison p-values are indicated.

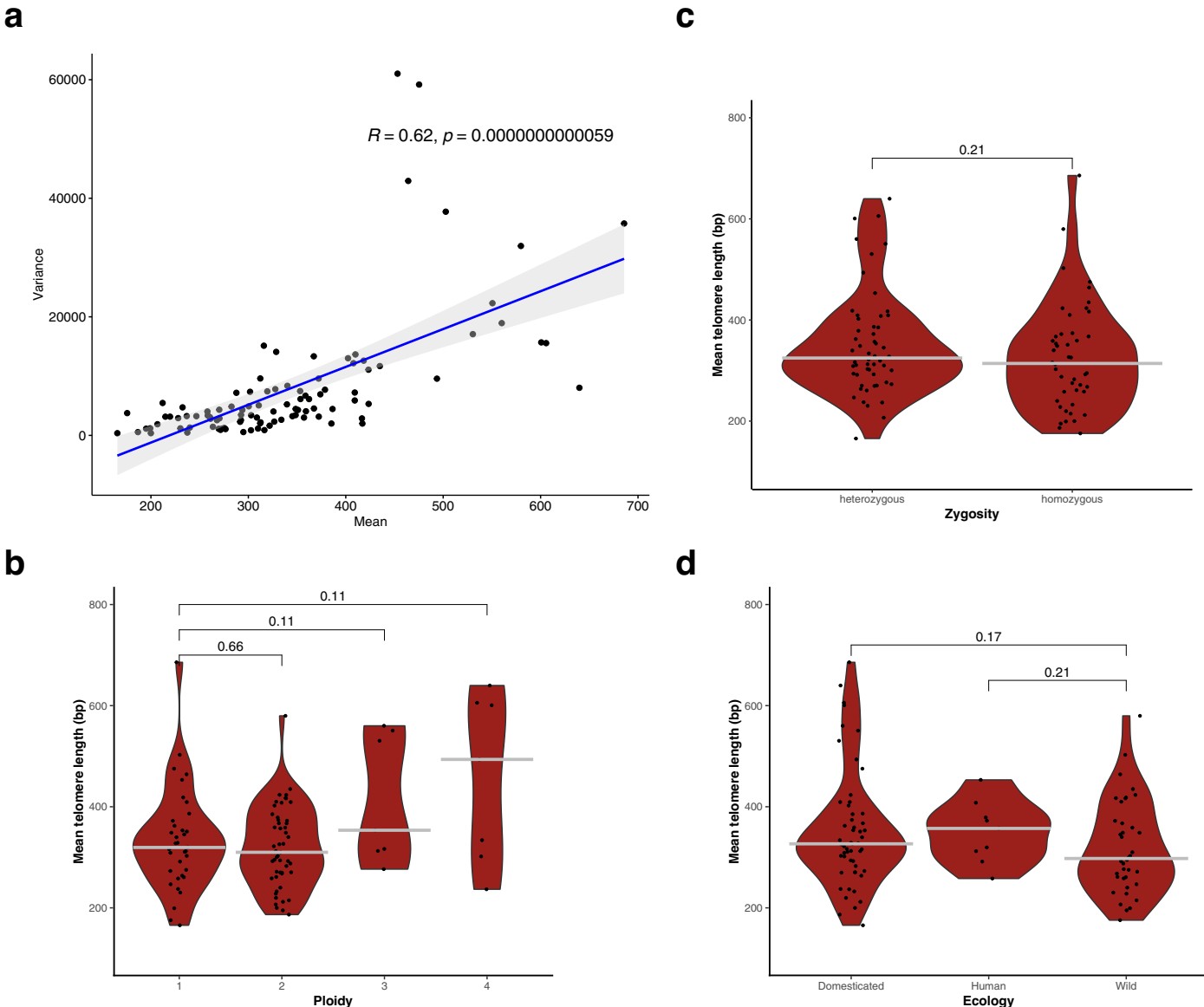

**Extended Data Fig. 5 | Telomere length properties. a**. Scatter plot showing the positive correlation between the mean telomere length and its variance per strain. The Pearson correlation coefficient and its associated p-value were calculated using the stat_cor method in R. The shaded error band represents the 95% confidence interval of the regression fit. Boxplots showing the distribution of telomere lengths per strain split by **b**. ploidy, **c**. zygosity and **d**. ecology. Median values are indicated by the gray crossbars in each violin plot. Two-sided Wilcoxon mean comparison p-values are indicated (**b**, **c** and **d**).

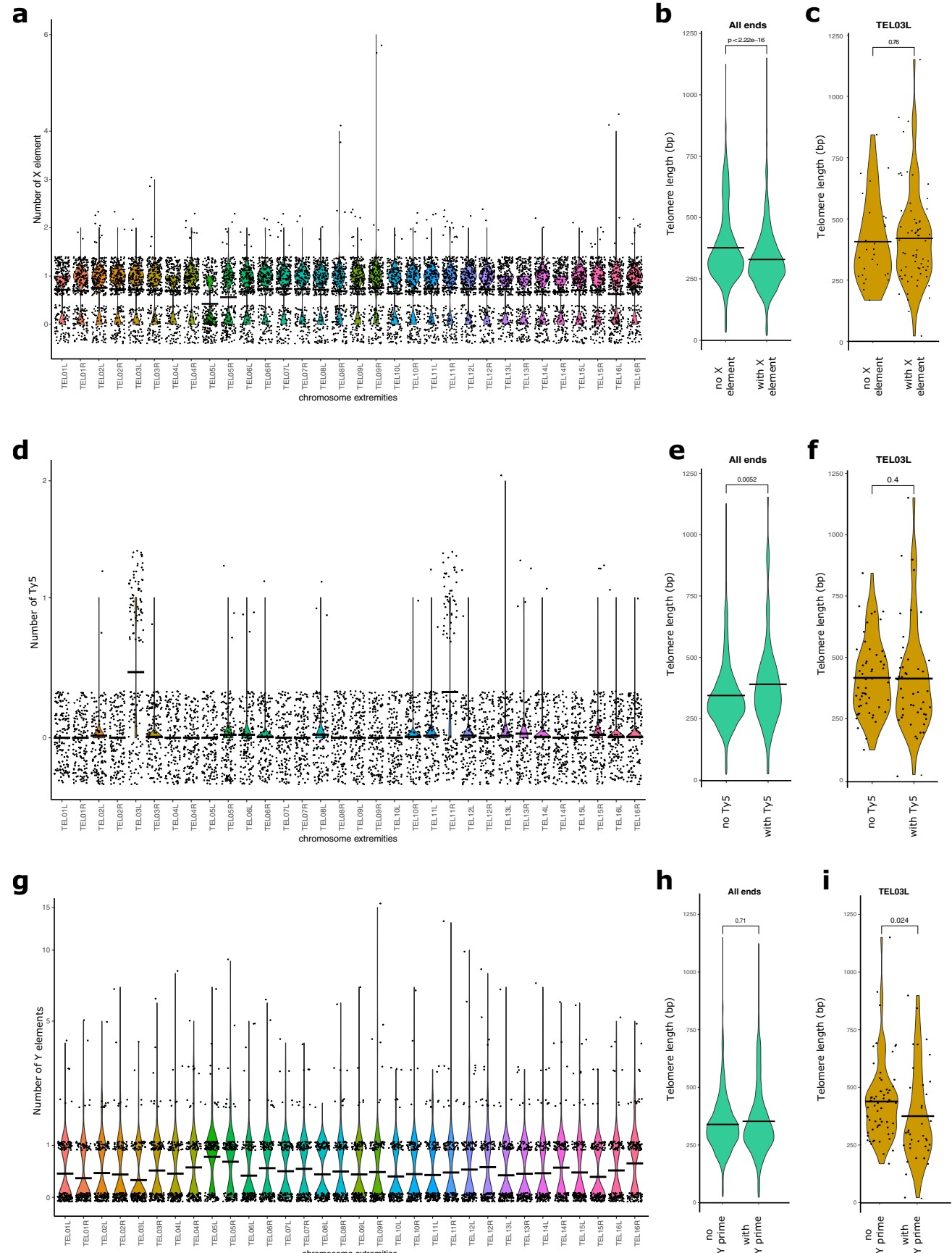

**Extended Data Fig. 6 | See next page for caption.**

**Extended Data Fig. 6 | Distribution of subtelomeric elements among the 100 de novo sequenced and assembled strains from the ScRAP. a**, **d** and **g** show the number of core X, Ty5 and Y' elements found on each chromosome end across the 100 strains, respectively. **b**, **e** and **h**. show the distributions of the mean telomere length in the presence or absence of the corresponding subtelomeric elements across all chromosome ends (n = 32 subtelomeres) and **c**, **f** and **i** across TELO3L (n = 32 subtelomeres). Median values are indicated by the black crossbars in each violin plot. Two-sided Wilcoxon mean comparison p-values are indicated (**b**, **c**, **e**, **f**, **h** and **i**).

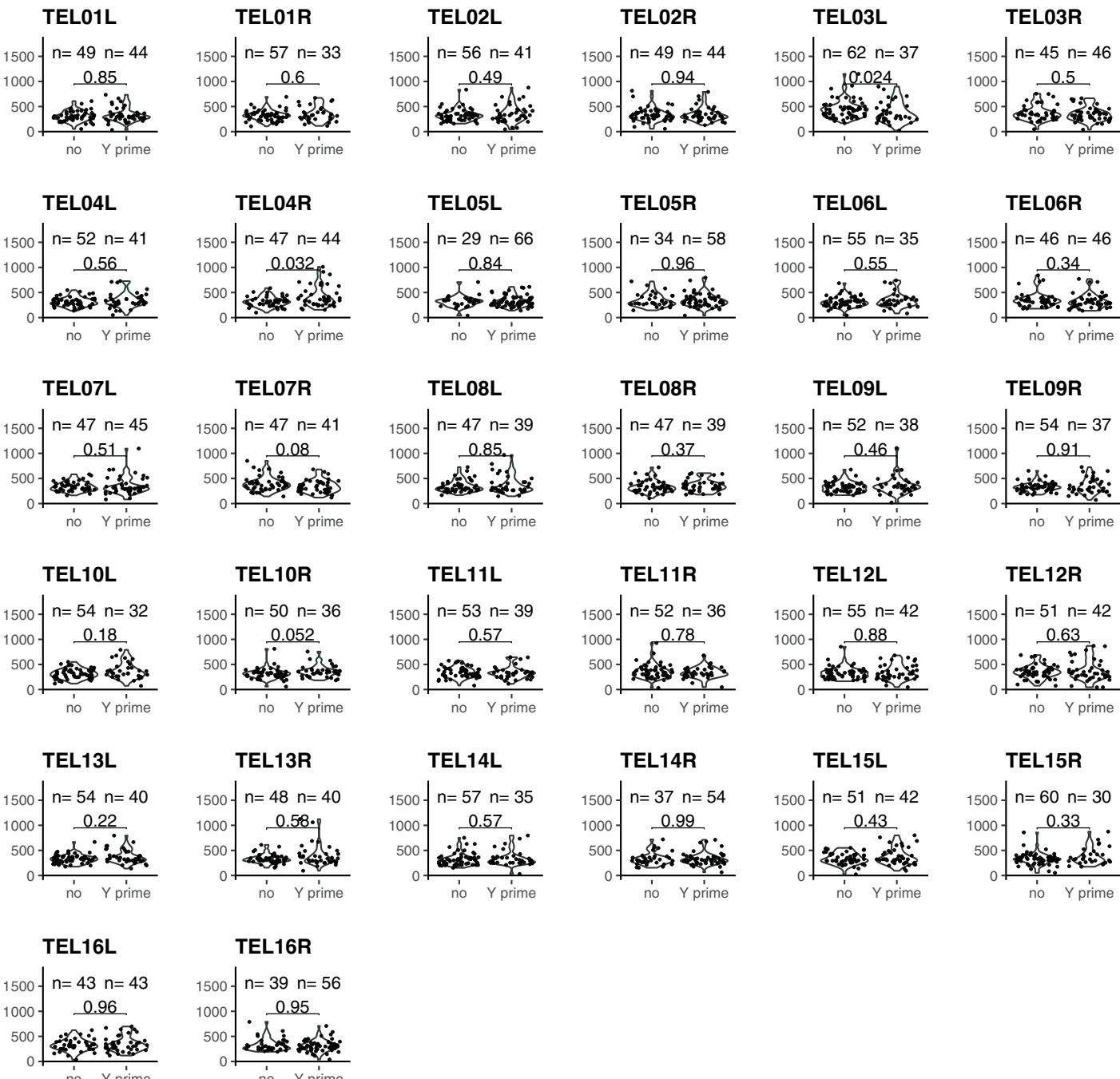

**Extended Data Fig. 7 | Distributions of the mean telomere length in the presence or absence of Y' elements for each chromosome end across the 100 de novo sequenced and assembled strains from the ScRAP.** Two-sided Wilcoxon mean comparison p-values are indicated. The number of telomeres that is used to derive each violin plot is indicated on the top of each individual plot.

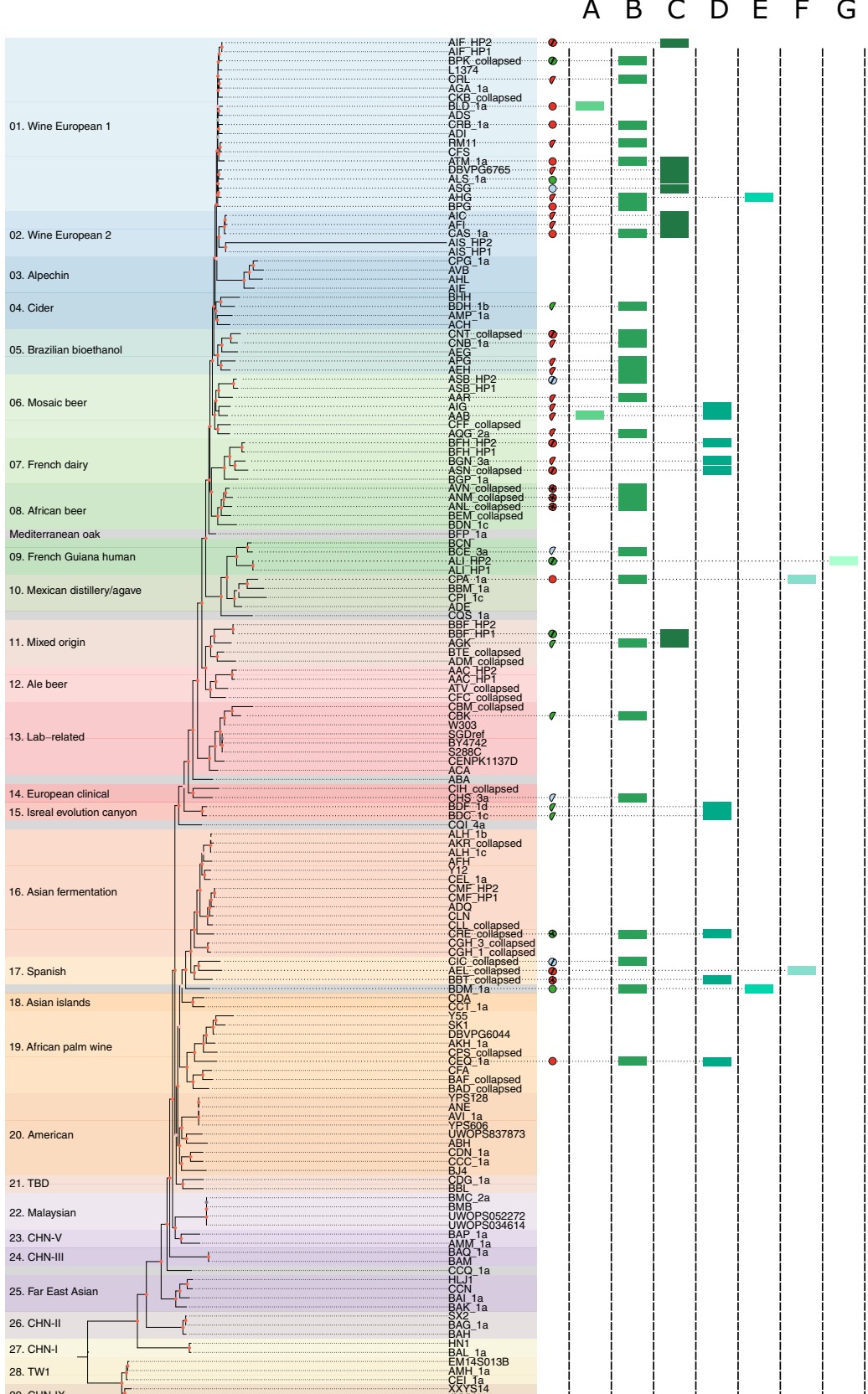

**Extended Data Fig. 8 | Distribution of the HGT regions in the ScRAP.** The phylogenetic tree on the left is identical to Fig. 1 and corresponds to a tree based on the concatenated protein sequence alignment of 1,612 1:1 ortholog. The green, red, blue and yellow symbols indicate the ecological origins. Ploidy levels and zygosity are symbolized by the shapes of the symbols as in Fig. 1.

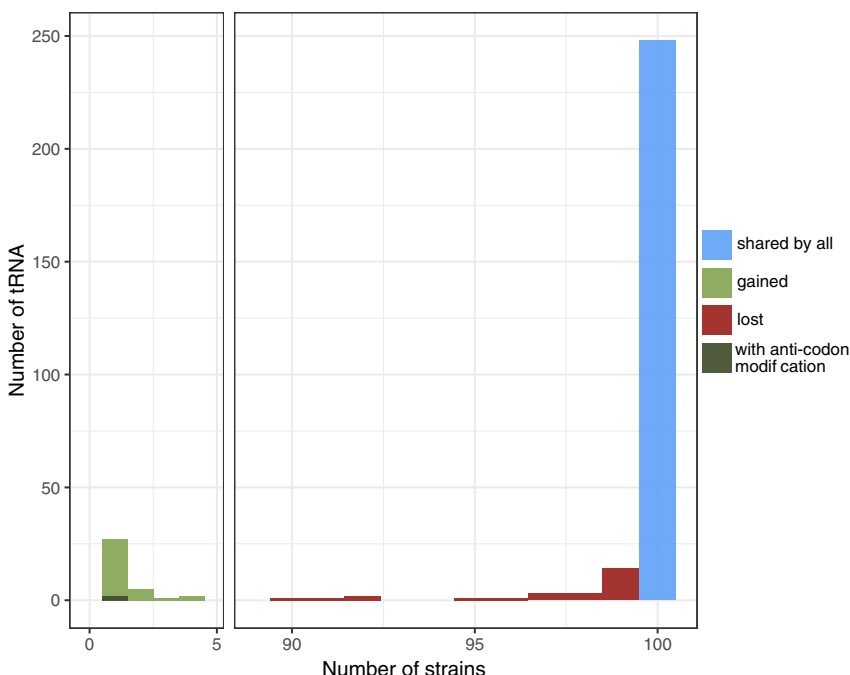

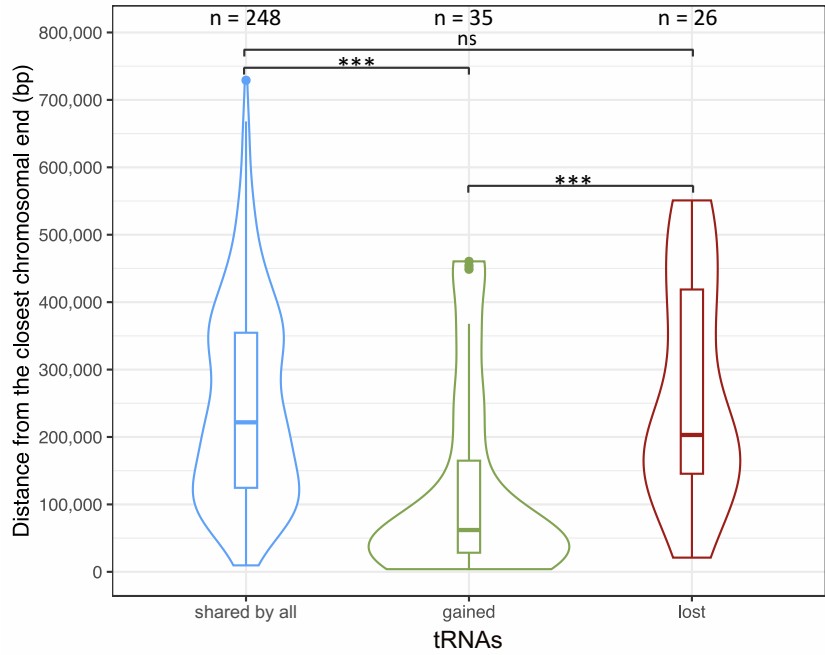

**Extended Data Fig. 9 | Distribution of the tRNA gene families in the ScRAP.**
**a**. Conservation of tRNA gene families across 100 isolates. **b**. Relative chromosomal location of conserved, gained and lost tDNA gene families. The horizontal lines in the boxplots correspond to the median, the lower and upper hinges correspond to the first and third quartiles and the whiskers extend up to 1.5 times the interquartile range. The three stars indicate P values < 0.01 from a two-sided Wilcoxon test (P = 3.707e-07 for shared by all vs gained and P = 0.0001122 for gain vs lost) and ns stands for non-significant (P = 0.4044 for shared by all vs lost).

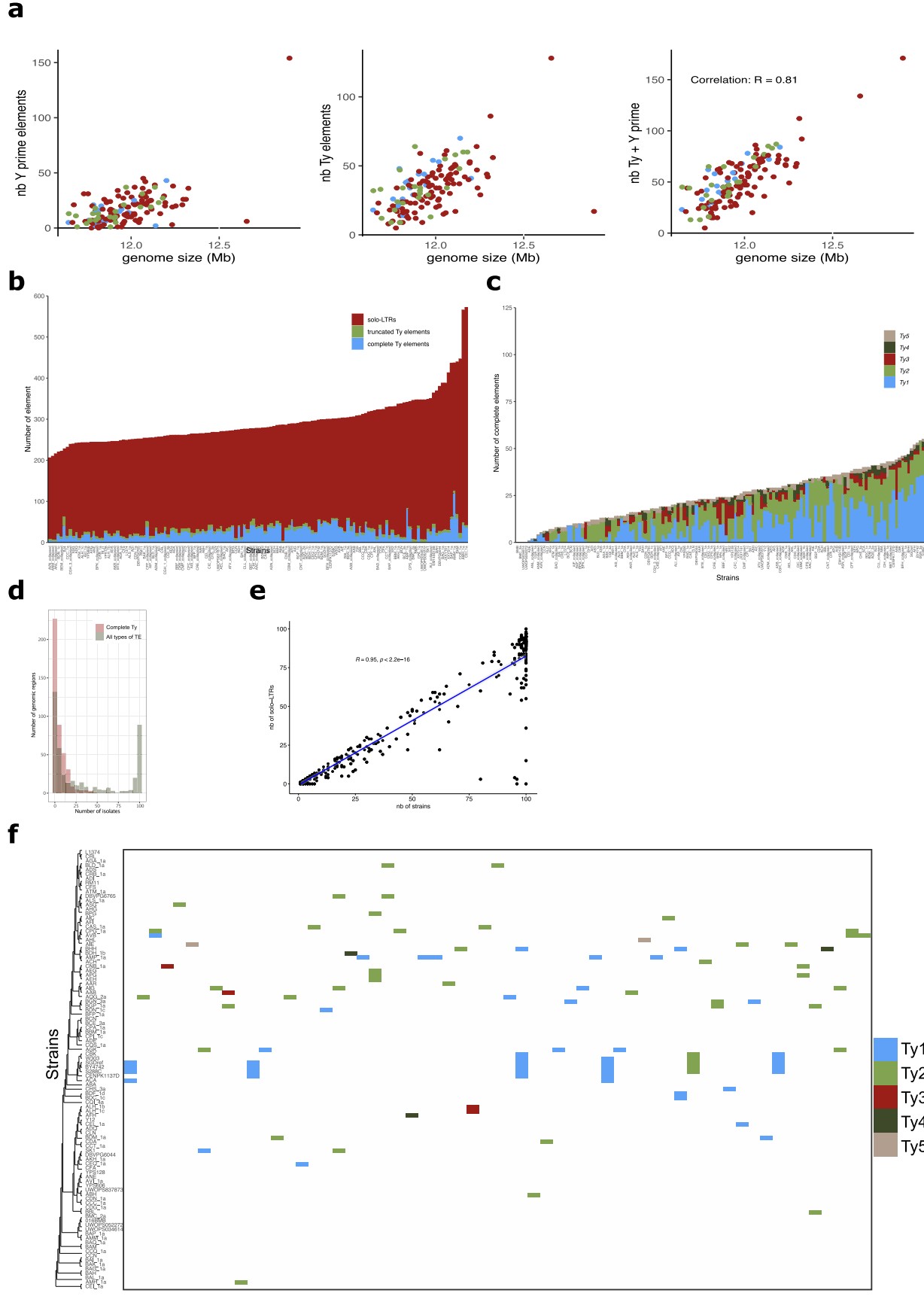

**Extended Data Fig. 10 | See next page for caption.**

**Extended Data Fig. 10 | Genome size variation in the ScRAP. a**. Scatter plots showing the genome size of each strain, split by dataset, as a function of the number of Y' elements (left), Ty elements (middle and Y' + Ty elements (right). **b**. Number of TE sequences per strain across the 142 haploid/collapsed genome assemblies. All sequences from the 5 Ty families are pooled together by category. **c**. Number of complete Ty elements per strain across the 142 haploid/collapsed genome assemblies. **d**. Distribution of the 126 insertion sites across the 100 haploid or homozygous genomes considering either the complete Ty elements or all types of TE sequences (complete, truncated and solo-LTRs). **e**. Scatter plot between number of solo-LTRs per insertion site and the number of strains sharing an insertion site. The Pearson correlation coefficient and its associated two-tailed t-test p-value were calculated using the stat_cor method in R. **f**. Map of the *de novo* insertions of complete Ty elements across the 100 homozygous explored strains. The map shows the 61 insertion sites in which only complete elements are found and never soloLTRs, which strongly suggests that these sites correspond to recent insertions. Strains are organized according to the nuclear phylogenetic tree (Fig. 1).

# Reporting Summary

## Statistics

For all statistical analyses, confirm that the following items are present in the figure legend, table legend, main text, or Methods section.

| n/a | Confirmed | |
|---|---|---|
| ☐ | ☒ | The exact sample size (*n*) for each experimental group/condition, given as a discrete number and unit of measurement |
| ☐ | ☒ | A statement on whether measurements were taken from distinct samples or whether the same sample was measured repeatedly |
| ☐ | ☒ | The statistical test(s) used AND whether they are one- or two-sided<br>*Only common tests should be described solely by name; describe more complex techniques in the Methods section.* |
| ☒ | ☐ | A description of all covariates tested |
| ☒ | ☐ | A description of any assumptions or corrections, such as tests of normality and adjustment for multiple comparisons |
| ☐ | ☒ | A full description of the statistical parameters including central tendency (e.g. means) or other basic estimates (e.g. regression coefficient) AND variation (e.g. standard deviation) or associated estimates of uncertainty (e.g. confidence intervals) |
| ☐ | ☒ | For null hypothesis testing, the test statistic (e.g. *F*, *t*, *r*) with confidence intervals, effect sizes, degrees of freedom and *P* value noted<br>*Give P values as exact values whenever suitable.* |
| ☒ | ☐ | For Bayesian analysis, information on the choice of priors and Markov chain Monte Carlo settings |
| ☒ | ☐ | For hierarchical and complex designs, identification of the appropriate level for tests and full reporting of outcomes |
| ☐ | ☒ | Estimates of effect sizes (e.g. Cohen's *d*, Pearson's *r*), indicating how they were calculated |

*Our web collection on statistics for biologists contains articles on many of the points above.*

## Software and code

Policy information about availability of computer code

| Data collection | Data collection:<br>Illumina paired end (HiSeq2500) and Oxford Nanopore Technology reads (Minion or Promethion) were generated for 100 strains. All other data was publicly available as indicated in the methods. |
|---|---|
| Data analysis | All custom scripts and softwares used in the study are publicly available for download:<br>Telofinder: https://github.com/GillesFischerSorbonne/telofinder<br>Script for aneuploidy detction: https://github.com/SAMtoBAM/aneuploidy_detection<br>Script for converting the vcf file into the fasta format: https://github.com/edgardomortiz/vcf2phylip<br>Script for down-sampling paired-end reads with seqtk: https://github.com/lh3/seqtk<br>Script for generating the non-redundant SV dataset: https://github.com/SAMtoBAM/MUMandCo/tree/master/nonredundant_population_datasets<br>Script for haplotype phasing pipeline is available at https://github.com/SAMtoBAM/PhasedDiploidGenomeAssemblyPipeline<br><br>The version number of the software/tools that we used are the following:<br>ADMIXTURE (version: 1.3.0)<br>A5-miseq (version: 20160825)<br>Guppy (version: 3.4.5)<br>Porechop (version: 0.2.4)<br>Picopore (version: 1.2.0)<br>ont-fast5-api (version: 0.3.2)<br>LRSDAY (version: 1.6.0) |

March 2021

Canu (version: 2.0)
SMARTdenovo (version: 5cc1356)
Racon (version: 1.4.7)
Medaka (version: 0.8.1)
Pilon (version: 1.23)
Ragout (version: 2.2)
Filtlong (version: v0.2.0)
Mummer4 (version: 4.0.0beta2)
Samtools (version: 1.11)
Gap5 (version: 1.2.14)
Minimap2 (version: 2.17)
bwa (version: 0.7.17)
GATK3 (version: 3.6-6)
GATK4 (version: 4.1.8.1)
GFF3toolkit (version: 2.1.0)
WhatsHap (version: 1.0)
NGMLR (version: 0.2.7 )
Sniffles (version: 2.0.2 )
nPhase (version: 1.1.3)
Hapo-G ( version: 1 )
Seqtk (version: 1.3-r106)
ncbi-BLAST (version: 2.2.31)
RM-BLAST (version: 2.2.28)
Circlator (version: 1.5.5)
MUM&Co (version: 3.8)
paftools (version: 2.17 )
bedtools (version: 2.27.1)
Tablet (version: 1.21.0.08 )
exonerate(version: 2.2.0)
Maker3 (version: 3.00.0-beta)
EVM (version: 1.1.1)
tRNAscan (version: 1.3.1)
RepeatMasker (version: open-4.0.7)
REannotate (version: 17.03.2015-LongQueryName)
BLAT (version: 36x8)
Proteinortho (version: 5.16b)
Mfannot (version: 1.35)
R (version: 3.5.1)
MACSE (version: 2.04)
IQtree (version: 1.6.12)
vcf2phylip (version: 2.8)
MAFFT (version: 7.471)
Newick Utilities (version: 1.6.0)
ggtree (version: 3.2.1)
phytools (version: 1.0-3)
TreeDist (version: 2.4.1)
MEGA (version: 11.0)
ClipKIT (version: ccc8bf)

For manuscripts utilizing custom algorithms or software that are central to the research but not yet described in published literature, software must be made available to editors and reviewers. We strongly encourage code deposition in a community repository (e.g. GitHub). See the Nature Portfolio guidelines for submitting code & software for further information.

# Data

Policy information about availability of data

All manuscripts must include a data availability statement. This statement should provide the following information, where applicable:

- Accession codes, unique identifiers, or web links for publicly available datasets
- A description of any restrictions on data availability
- For clinical datasets or third party data, please ensure that the statement adheres to our policy

All sequencing data and assembly/annotation files were deposited in the European Nucleotide Archive (https://www.ebi.ac.uk/ena/browser/home) under the umbrella project PRJEB59869. The project accession for the raw sequencing data (fast5, nanopore fastq, illumina fastq) is PRJEB50706. The assembly/annotation accessions are PRJEB59413, PRJEB59129, PRJEB59231, PRJEB59232, PRJEB59230 for unphased nuclear, haplotype 1 (HP1), haplotype 2 (HP2), haplotype (HP, for polyploids) and mitochondrial assemblies, respectively. Each accession for individual assemblies is indicated in Supp. Table 1 (nuclear) and Supp. Table 3 (mitochondrial).

# Human research participants

Policy information about studies involving human research participants and Sex and Gender in Research.

| | |
|---|---|
| Reporting on sex and gender | NA |
| Population characteristics | NA |
| Recruitment | NA |
| Ethics oversight | NA |

Note that full information on the approval of the study protocol must also be provided in the manuscript.

# Field-specific reporting

Please select the one below that is the best fit for your research. If you are not sure, read the appropriate sections before making your selection.

☐ Life sciences ☐ Behavioural & social sciences ☒ Ecological, evolutionary & environmental sciences

For a reference copy of the document with all sections, see nature.com/documents/nr-reporting-summary-flat.pdf

# Ecological, evolutionary & environmental sciences study design

All studies must disclose on these points even when the disclosure is negative.

| | |
|---|---|
| Study description | This study includes 100 newly sequenced genomes (ii) 18 re-assembled genomes and (iii) 24 publically available assemblies. Overall, the study consists of 142 haploid or collapsed assemblies (one per strain), 55 haplotype-resolved assemblies comprising two phased assemblies per heterozygous diploid (21 strains) and one haplo-phased assembly per heterozygous polyploid (13 strains), totaling 197 nuclear genome assemblies. The study also contains 136 mitochondrial chromosome assemblies (114 de novo and 22 publicly available). |
| Research sample | The study includes 142 Saccharomyces cerevisiae strains. |
| Sampling strategy | The strains were sampled to represent the species' phylogenetic and ecological diversity, with varying ploidy and heterozygosity levels. |
| Data collection | Strains were recovered from culture collections. |
| Timing and spatial scale | NA |
| Data exclusions | NA |
| Reproducibility | All genomic assemblies were independently performed twice with 2 different algorithms (Canu and SMARTdenovo) |
| Randomization | NA |
| Blinding | Blinding was not relevant because each sequenced strain was unique. |

Did the study involve field work? ☐ Yes ☒ No

# Reporting for specific materials, systems and methods

We require information from authors about some types of materials, experimental systems and methods used in many studies. Here, indicate whether each material, system or method listed is relevant to your study. If you are not sure if a list item applies to your research, read the appropriate section before selecting a response.

## Materials & experimental systems

| n/a | Involved in the study |
|---|---|
| ☒ ☐ | Antibodies |
| ☒ ☐ | Eukaryotic cell lines |
| ☒ ☐ | Palaeontology and archaeology |
| ☒ ☐ | Animals and other organisms |
| ☒ ☐ | Clinical data |
| ☒ ☐ | Dual use research of concern |

## Methods

| n/a | Involved in the study |
|---|---|
| ☒ ☐ | ChIP-seq |
| ☒ ☐ | Flow cytometry |
| ☒ ☐ | MRI-based neuroimaging |

