## [Peer Review File · Nature Genetics]

Peer Review Information

Manuscript Title: Telomere-to-telomere assemblies of 142 strains characterize the genome structural landscape in *Saccharomyces cerevisiae*

Corresponding author name(s): Professor Joseph Schacherer, Gianni Liti, Dr Gilles Fischer

Reviewer Comments & Decisions:

Decision Letter, initial version:

5th Jan 2023

Dear Dr Fischer,

Your Article, "142 telomere-to-telomere assemblies reveal the genome structural landscape in *Saccharomyces cerevisiae*" has now been seen by 3 referees. You will see from their comments below that while they find your work of interest, some important points are raised. We are interested in the possibility of publishing your study in Nature Genetics, but would like to consider your response to these concerns in the form of a revised manuscript before we make a final decision on publication.

We therefore invite you to revise your manuscript taking into account all reviewer and editor comments. Please highlight all changes in the manuscript text file. At this stage we will need you to upload a copy of the manuscript in MS Word .docx or similar editable format.

*2) If you have not done so already please begin to revise your manuscript so that it conforms to our Article format instructions, available [here](http://www.nature.com/ng/authors/article_types/index.html). Refer also to any guidelines provided in this letter.

*3) Include a revised version of required Reporting Summary and Editorial Policy Checklist (**Please note that we will NOT be able to move forward with your manuscript without the required Reporting Summary and Editorial Policy Checklist.**):

-- Reporting summary: <https://www.nature.com/documents/nr-reporting-summary.pdf>

-- Editorial policy checklist: <https://www.nature.com/documents/nr-editorial-policy-checklist.pdf>

[redacted]

We hope to receive your revised manuscript within 3 to 6 months. If you cannot send it within this time, please let us know.

Sincerely,
Wei

Wei Li, PhD
Senior Editor
Nature Genetics
New York, NY 10004, USA
www.nature.com/ng

Reviewers' Comments:

Reviewer #1:

Remarks to the Author:

O'Donnell et al report an extensive and systematic analysis of structural variation (SV) in *Saccharomyces cerevisiae*. It is a very innovative and exciting manuscript because it does not just present a dull catalog of SVs. Instead, the authors have dug deeply into the data and they bring a lot of new biological and evolutionary insight to the patterns of SV that they observe. They also took a broad view of SV, considering aspects such as telomere length and structures, and patterns of variation in the locations of mobile genetic elements, which (to my knowledge) have not often been considered in analyses of SV before. Moreover, the manuscript is based on a very solid and carefully chosen dataset: gapless genome sequences from 142 *S. cerevisiae* strains representing the complete phylogenetic diversity of this species and also its natural diversity of ploidy and heterozygosity. In my opinion, this study brings the analysis of SV in *S. cerevisiae* to a higher level of detailed knowledge than has been achieved in any other organism, including humans. It is a veritable tour de force from the 3 co-PIs and their team.

There are many new 'stories' in this manuscript. For me, the highlights were (1) quantification of the effects of structural rearrangement on transcription of the neighboring genes (Fig 2I); (2) discovery that horizontal gene transfer (HGT) from other yeast species at the ends of chromosomes can generate new telomeres with structures that transition from the donor species' repeat pattern to the *S. cerevisiae* repeat pattern (Fig 5C); (3) discovery of HGT of two complete chromosomes from *S. kudriavzevii* into one strain of *S. cerevisiae* (Fig 5E); (4) discovery that, like protein-coding genes, the repertoire of tRNA genes in a genome undergoes duplication, deletion, and gain (Fig 5F). Each of these stories is a new discovery: it is not just more-detailed view of something we already knew about, it is something the field had previously not thought about, or had no data about.

The manuscript has been prepared to a very professional standard. I have several comments and questions, but no major criticisms.

L37. Mention that some of the ScRAP strains are phased polyploids. Resolving and analyzing the structure of these genomes is a major achievement of this manuscript, so highlight it in the Abstract.

L113 and L1086. The number of one-to-one orthologs (1,618) is surprisingly low, considering that there are ~5700 protein-coding genes in the reference strain. Any comment on why it is so low?

L171. Begin this section by mentioning the size limits you used to define SVs (e.g. indels were required to be > 50 bp). These definitions are in the Fig 2 legend, but I think they should be in the text because they are important, because the number of SVs detected is strongly dependent on how they are defined.

L664-669. Related to the previous point, do these comparisons of SV density in other species all use the same definitions for the minimum size of SVs (as in Fig 2 legend?). If not, the densities are not directly comparable.

Figure 2 and the text on L169-327 is a very informative summary of the properties of SVs, but I would have liked to see some more information about any SVs that have a high Minor Allele Frequency (Fig 2D) and so might be maintained by balancing selection. This could be done by adding a MAF column to Table S5.

Related to the previous point, I looked at the data in Table S5 for an SV that I know well, the 25-kb ‘flip/flop’ inversion on chromosome XIV (row 4059 of the table) which is an SV with a high MAF. The analysis seems to have captured this SV correctly, but I’m puzzled because Column B for this SV contains the names of 251 genomes (or haplotypes) that contain this SV. It made me realize that I don’t know the number of genomes/haplotypes that were analyzed to make this table (so I also can’t calculate the MAF). The number is obviously much higher than the number 197 given in Table 1 and L95-99, and I don’t know why.

L271-274 and L286-288. When commenting on the intra-genic SVs and gene fusions, it would be of interest to know what percentage are in-frame (and so make a protein fusion) as opposed to out-of-frame (and so truncate the protein). I realize that it might be difficult to calculate these percentages, but if it can be done easily please include them. I think it’s a bit misleading to call an SV a “gene fusion” if it isn’t actually in the frame necessary to make a chimeric protein.

L329-L348. Figure 3B is interesting, but I don’t think Figure 3A and the text on L329-348 are interesting. I don’t find it surprising that a tree constructed from a presence/absence matrix of SVs is similar to, but less accurate than, a tree drawn by standard SNP or amino acid sequence analysis, and I don’t really see the point of it. So I suggest moving Fig 3A and this text to Supp Info.

L419 and L41 (Abstract). Fig 4C shows that larger chromosomes are more often involved in complex aneuploidies than smaller ones. Is this not a trivial result – exactly what we would expect if rearrangement points were distributed randomly, i.e. big chromosomes have more potential to be

involved in these events because they contain more DNA? So I don't see why you highlight this result in the Abstract. If I have missed something, please explain in more detail why this is not an expected result.

Minor suggestions, typos, grammar, etc:

Table S1 says "Public assembly" for some genomes that were analyzed but not sequenced by the authors. Please give database accession numbers for these.

L42. Change classically to typically

L50. Change to "... remain unassembled with older technologies."

L114 and L147. There aren't 23 other *Saccharomyces* species, so I guess you mean 23 strains from other *Saccharomyces* species.

L127. For clarity, change to "sister haplotypes (HP1, HP2)"

L142 Replace the "Green, red, blue..." sentence with "Colors and shapes are used as a key to strain origin and ploidy". The meaning of the "/" symbol in the cell for human heterozygous diploids in Fig 1A is not explained.

L159. Change splitted to split.

L166. Change his to its.

L214. Has the X-axis in Figure 2F been truncated at 10 kb? If so, say this in the legend.

L232. Change "but" to "except".

L291. I didn't understand the connection between the two parts of this sentence. Consider changing "while it is generally assumed" to "in contrast to the general assumption..." if that's what you mean.

L349. Say here that the numbers of SVs and SNVs/indels are calculated relative to the reference genome (S288C), if that's correct.

L403. There seems to be a mistake in the diagram of ASG in Fig 4A. For chromosome II, 4 copies are drawn but the karyotype is written as $2n + 1 \cdot \text{chrII}$.

L535-547. The AIS genome (Fig 5E) is very interesting. Does it contain complete rDNA units from both *Sk* and *Sc*?

L648. Change to “clade 13 (lab-related) in which...”

L671. I’m not sure what “that would” means in the sentence beginning “A genomic clock that would synchronize...”. The sentence seems to be grammatically incorrect, and I’m not sure if this clock is a hypothetical idea or a real observation. Maybe change it to “synchronizes” or “has been postulated to synchronize” ?

Signed review by Ken Wolfe

Reviewer #2:

Remarks to the Author:

The manuscript by O'Donnell et al. describes complete genome assemblies for 142 representative strains of the model yeast *Saccharomyces cerevisiae*. Using these assemblies, the authors characterize the structural variation of the species in great detail.

The analyses are generally well done and clearly described, although the (very broad) scope of the data and the in-depth analyses means that there are a few places where it was hard for me to figure out what the authors did or meant (at least I couldn't after 2-3 reads). Here are a few cases:

Lines 97-98 and Table 1: it is not clear to me why the authors do not provide the haplotype phased assemblies for the 13 polyploid strains. As far as I understand, the authors provide 13 genome assemblies, one for each strain, rather than 46 assemblies (6×3 for the 6 triploid strains and 7×4 for the 7 tetraploid strains)

Lines 99-101: Not clear why only 135 mitochondrial genomes are available from 142 genomes. Can you briefly add an explanation?

Section beginning on line 328: what are SNVs and how they differ from SNPs? (they are never defined - I presume they both refer to single nucleotide polymorphisms)

Lines 344-346: The authors infer that there is lower information / signal on the SV tree than on the SNP tree but then conclude that the signal from the two types of data is complementary. I am not sure I understand how could this be, especially if one considers a specific genomic region that contains both SNPs and SVs (I would expect that patterns of variation in both types of data would reflect the same evolutionary history).

In a few places, the authors use over-the-top language that should be toned down. Here are a few cases:

Lines 102-104: Providing a brief summary of the contiguity and completeness data would have been better than telling us that the dataset is "unprecedented"

Various lines: "first rigorous" (line 77), "unprecedented" (line 102), "notable improvement" (line 115), etc. In my opinion these should be all removed and the authors should let the data speak for themselves...

Line 126: the title of this section is "SVs impact gene expression" - given the subtle effect, it would be better to say that "SVs can impact gene expression" since only a small fraction of SVs are shown to affect gene activity.

I believe that all these points are straightforward to address and that the dataset provided will be of high value to the community. If I had one major concern with the manuscript, this would be the lack of a clear take-home message or conclusion. I understand that the authors have characterized structural variation of a yeast species in unprecedented detail, but what does this tell us? I see lots of detailed characterizations of the different types of events that can occur and the frequencies with which they occur, but I am missing the take-home message. Tellingly, there are ~20+ pages of results and only 1 page of discussion. Maybe that's a whole separate study, but I think the authors are missing here a great opportunity to state the broader relevance of their results and findings for population biology and genome evolution.

Minor points:

Line 567: "in less than" -> "in fewer than"

Line 268: "majoritarily" -> "mostly"

Lines 670-673: check this sentence and rephrase - it doesn't quite make sense

Reviewer #3:

Remarks to the Author:

The authors generate the *Saccharomyces cerevisiae* Reference Assembly Panel (ScRAP) comprising telomere-to-telomere genome assemblies for 142 strains from various geographical and ecological distributions. The ScRAP is important for population genomics studies of *S. cerevisiae*, in terms of phylogeny, ploidy, heterozygosity and ecological origins. Using the generated ScRAP dataset, the authors characterize the structural variant (SV) and reveal valuable findings regarding the evolutionary dynamics of complex regions. The high quality of data, the precise presentations, as well as the appropriate usage of statistics, are all satisfactory. The conclusions are reliable. The manuscript is also well written. In general, I suggest to accept this manuscript.

There are only some minor issues:

1. In Page 5, L98 states 13 strains are heterozygous polyploidy, which should include 6 triploid and 7 tetraploid as shown in Table 1. However, the number of triploid is 7 in figure 1A and 1C. Please check and confirm the information.
2. It is interesting that all of the triploid and tetraploid (13 or 14 strains in total) in this study are isolated from fermentations, especially from wine and beer. I wonder if the authors find anything new in the structure variants analysis that may help to explain the connection between multi-ploidy and fermentation ability (or special food flavor)?

Author Rebuttal to Initial comments

Dear editor and reviewers,

We would like to thank you for your time in reviewing this manuscript. We appreciated the overall enthusiasm of the reviews and their constructive feedback. In addition to the changes suggested by the reviewers, we have reworded some parts to improve clarity and shorten the manuscript. All changes have been saved using tracked changes in word. Point-by-point responses to reviewer comments are provided below.

We hope that our work is now suitable for publication in Nature Genetics.

Author response:

Reviewer #1:

Remarks to the Author:

O'Donnell et al report an extensive and systematic analysis of structural variation (SV) in *Saccharomyces cerevisiae*. It is a very innovative and exciting manuscript because it does not just present a dull catalog of SVs. Instead, the authors have dug deeply into the data and they bring a lot of new biological and evolutionary insight to the patterns of SV that they observe. They also took a broad view of SV, considering aspects such as telomere length and structures, and patterns of variation in the locations of mobile genetic elements, which (to my knowledge) have not often been considered in analyses of SV before. Moreover, the manuscript is based on a very solid and carefully chosen dataset: gapless genome sequences from 142 *S. cerevisiae* strains representing the complete phylogenetic diversity of this species and also its natural diversity of ploidy and heterozygosity. In my opinion, this study brings the analysis of SV in *S. cerevisiae* to a higher level of detailed knowledge than has been achieved in any other organism, including humans. It is a veritable tour de force from the 3 co-PIs and their team.

There are many new 'stories' in this manuscript. For me, the highlights were (1) quantification of the effects of structural rearrangement on transcription of the neighboring genes (Fig 2I); (2) discovery that horizontal gene transfer (HGT) from other yeast species at the ends of chromosomes can generate new telomeres with structures that transition from the donor species' repeat pattern to the *S. cerevisiae* repeat pattern (Fig 5C); (3) discovery of HGT of two complete chromosomes from *S. kudriavzevii* into one strain of *S. cerevisiae* (Fig 5E); (4) discovery that, like protein-coding genes, the repertoire of tRNA genes in a genome undergoes duplication, deletion, and gain (Fig 5F). Each of these stories is a new discovery: it is not just more-detailed view of something we already knew about, it is something the field had previously not thought about, or had no data about.

We would like to thank the reviewer for his highly positive opinion on our work.

The manuscript has been prepared to a very professional standard. I have several comments and questions, but no major criticisms.

L37. Mention that some of the ScRAP strains are phased polyploids. Resolving and analyzing the

structure of these genomes is a major achievement of this manuscript, so highlight it in the Abstract.

We added a sentence to highlight this in the abstract and also in the first paragraph of the result section.

L113 and L1086. The number of one-to-one orthologs (1,618) is surprisingly low, considering that there are ~5700 protein-coding genes in the reference strain. Any comment on why it is so low?

We agree with the reviewer that this number is very low and does not faithfully represent the complete set of orthologs across the ScRAP. We did not investigate this problem in-depth because the set of 1,618 orthologs was sufficiently informative to reconstruct a reliable phylogeny. This set of orthologous genes was computed by *proteinortho* (Lechner et al. 2011 BMC Bioinformatics) based on the annotated CDSs across 142 *S. cerevisiae* strains and 23 outgroup species. Therefore, a single genome with a lower annotation quality (possibly an outgroup species) would be sufficient to drastically reduce the set of 1:1 orthologs. Additionally, orthologous genes with occasional duplication or loss in any of our selected strains will be excluded, which also contribute to decreasing the number. Such impact will be further magnified given that we intentionally sampled strains based on their SV and gene content diversity.

L171. Begin this section by mentioning the size limits you used to define SVs (e.g. indels were required to be > 50 bp). These definitions are in the Fig 2 legend, but I think they should be in the text because they are important, because the number of SVs detected is strongly dependent on how they are defined.

We added the following sentence: “These calls consist of CNVs such as deletions, insertions, duplications and contractions of repeated sequences (> 50 bp) and copy-neutral rearrangements including inversions (> 1 kb) and translocations (> 10 kb).”

L664-669. Related to the previous point, do these comparisons of SV density in other species all use the same definitions for the minimum size of SVs (as in Fig 2 legend?). If not, the densities are not directly comparable.

We thank the reviewer for this thoughtful remark. The same 50 bp cutoff was used to detect SVs in all other species except *Drosophila* where the authors focused on SVs > 100 bp and limited to euchromatic regions. We added the following note in the main text: “Note that the density is likely underestimated in *Drosophila* because only large (> 100 bp as compared to > 50 bp in all other species) SVs limited to euchromatic regions were considered.”

Figure 2 and the text on L169-327 is a very informative summary of the properties of SVs, but I would have liked to see some more information about any SVs that have a high Minor Allele Frequency (Fig 2D) and so might be maintained by balancing selection. This could be done by adding a MAF column to Table S5.

We thank the reviewer for this suggestion. We added an “allele_frequency” column to Table S5 (column H) that corresponds to the frequency of strains carrying the SV relative to the reference genome (#strains in column C divided by 141). We think that AF is easier to interpret than MAF given that all SVs are defined relative to the reference genome.

Related to the previous point, I looked at the data in Table S5 for an SV that I know well, the 25-kb ‘flip/flop’ inversion on chromosome XIV (row 4059 of the table) which is an SV with a high MAF. The analysis seems to have captured this SV correctly, but I’m puzzled because Column B for this SV contains the names of 251 genomes (or haplotypes) that contain this SV. It made me realize that I don’t know the number of genomes/haplotypes that were analyzed to make this table (so I also can’t calculate the MAF). The number is obviously much higher than the number 197 given in Table 1 and L95-99, and I don’t know why.

We apologize for the lack of clarity. This column contains the assemblies that validated the SVs. The number of assemblies can exceed 197 because we used both the best and the runner up assemblies (as all strains were assembled by both Canu and SMARTdenovo (methods)) during the SV calling process. This strategy was used in order to avoid calling events only present in a single assembly due to assembly errors. In haploid and homozygous diploid/polyploid strains, this gives us a total of two possible assemblies, however this number is higher for heterozygous diploid/polyploid genomes. For example, for heterozygous diploids, in addition to the haplotype-collapsed assemblies from both Canu and SMARTdenovo, each haplotype assembly (HP1 and HP2) was also assembled by both Canu and SMARTdenovo, therefore in this case we used 6 genomes (2 unphased, 4 phased) per strain for SV calling. In total, there is a maximum of 459 possible allele assemblies per SV in this column. In order to simplify this information, we added a new column (column C) to Table S5 containing only the names of the strains carrying the SVs (the highest possible number of strains being 141). This new column C is used to calculate the AF presented in the new column H (see above).

L271-274 and L286-288. When commenting on the intra-genic SVs and gene fusions, it would be of interest to know what percentage are in-frame (and so make a protein fusion) as opposed to out-of-frame (and so truncate the protein). I realize that it might be difficult to calculate these percentages, but if it can be done easily please include them. I think it's a bit misleading to call an SV a "gene fusion" if it isn't actually in the frame necessary to make a chimeric protein.

We agree with the reviewer that this information would be very interesting. We are actually working on this problem, but as mentioned by the reviewer, it is not straightforward to access this information for all types of SVs in all strains. We changed the term 'gene fusion' into 'putative gene fusion' and added the precision that these events likely comprise both in-frame and out-of-frame fusions, in undetermined proportions. We have also added in the legend of Figure 2 that the proportions of in-frame and out-of-frame fusions have not been determined.

L329-L348. Figure 3B is interesting, but I don't think Figure 3A and the text on L329-348 are interesting. I don't find it surprising that a tree constructed from a presence/absence matrix of SVs is similar to, but less accurate than, a tree drawn by standard SNP or amino acid sequence analysis, and I don't really see the point of it. So I suggest moving Fig 3A and this text to Supp Info. We moved the Fig 3A and the corresponding text to Supp Info. We also moved the original panel C from Fig. 2 to a Supp. Fig. (now called Supp Fig. 6) and replaced it with the original Fig 3B.

L419 and L41 (Abstract). Fig 4C shows that larger chromosomes are more often involved in complex aneuploidies than smaller ones. Is this not a trivial result – exactly what we would expect if rearrangement points were distributed randomly, i.e. big chromosomes have more potential to be involved in these events because they contain more DNA? So I don't see why you highlight this result in the Abstract. If I have missed something, please explain in more detail why this is not an expected result.

We believe this is neither a trivial result nor an expected result, for several reasons. First, complex aneuploidies represent a new type of genetic diversity that has not been formally described before. Complex aneuploidy is different from previously reported cases of 'segmental aneuploidies' (Selmecki et al., 2006) in the sense that a centromere must be contained within the copy-number modified region to represent a complex aneuploidy. In humans, chromosome arm aneuploidies (CAAs) occur frequently in cancer and even more frequently than whole chromosome aneuploidies in certain cancer types (Shukla et al., 2020; Taylor et al., 2018). But these CAAs, although they have the potential to represent complex aneuploidies, have not been formally defined as centromere-containing segments, possibly due to the complex repetitiveness of human centromeres that prevents their assembly. Second, complex aneuploidy should be extremely rare as they are expected to occur at a rate corresponding to the product of the rates of aneuploidy and SV formation. However, they are widespread in the population, as the re-analysis of the genomes of

993 strains has shown that up to 18% of all aneuploidies could be complex, suggesting that they are possibly adaptive and selected for in response to harsh environments, as has been shown for simple aneuploidies (Mulla et al., 2014). Third, several studies on *Saccharomyces* reported a significant negative correlation between chromosome size and the rate of simple aneuploidy (Gilchrist and Stelkens, 2019). We observed the same negative correlation for the 379 cases of simple aneuploidies that we detected here by reanalyzing the 1,011 genome dataset (not shown). This is probably due to the fitness cost of extra chromosomes being proportional to the total number of genes present in the excess chromosomes. In line with this, CAAs frequency in human cancers is inversely related to arm length (Beroukhim et al., 2010). Strong fitness costs of large aneuploid chromosomes would therefore outweigh the benefits of increased dosage of most (if not all) genes located on large chromosomes, thereby restricting the adaptive role of aneuploidy to genes located on the small chromosomes. Here we show that larger chromosomes are more often involved in complex aneuploidies than smaller ones. Therefore, we propose that complex aneuploidies would open a specific adaptive route that would not be accessible to simple aneuploidies. Complex aneuploidies would reduce the fitness cost of excess large chromosomes by shortening them through deletions and/or translocations, thereby allowing the benefit of specific gene dosage increases to outweigh the fitness cost of the shortened aneuploidy chromosomes. In other words, complex aneuploidies would allow for an extension of the adaptive role of aneuploidy for genes located on large chromosomes.

Minor suggestions, typos, grammar, etc:

Table S1 says “Public assembly” for some genomes that were analyzed but not sequenced by the authors. Please give database accession numbers for these.

The database accessions have been added for public assemblies both in Supp. Table 1 (nuclear) and Supp. Table 3 (mitochondrial). Moreover, we also added all accessions for raw sequencing data and assemblies for all other strains in these tables.

L42. Change classically to typically

Done

L50. Change to “... remain unassembled with older technologies.”

Done

L114 and L147. There aren't 23 other *Saccharomyces* species, so I guess you mean 23 strains from other *Saccharomyces* species.

Yes, thank you for pointing to this mistake. We changed the sentence accordingly.

L127. For clarity, change to “sister haplotypes (HP1, HP2)”

Done

L142 Replace the “Green, red, blue...” sentence with “Colors and shapes are used as a key to strain origin and ploidy”.

Done

The meaning of the “/” symbol in the cell for human heterozygous diploids in Fig 1A is not explained.

The “/” is only there in order to split the circle into 2 halves because the key that is used to symbolize a heterozygous diploid is a split circle. The same is true for the other cells of the ‘heterozygous diploid’ column in Fig 1A.

L159. Change splitted to split.

Done

L166. Change his to its.

Done

L214. Has the X-axis in Figure 2F been truncated at 10 kb? If so, say this in the legend.

Yes, the X-axis has been truncated to 10 kb. We added this precision in the legend.

L232. Change "but" to "except".

Done

L291. I didn't understand the connection between the two parts of this sentence. Consider changing "while it is generally assumed" to "in contrast to the general assumption..." if that's what you mean.

We changed accordingly.

L349. Say here that the numbers of SVs and SNVs/indels are calculated relative to the reference genome (S288C), if that's correct.

This sentence has been added in the legend of the new Fig. 2C panel.

L403. There seems to be a mistake in the diagram of ASG in Fig 4A. For chromosome II, 4 copies are drawn but the karyotype is written as $2n + 1 \cdot \text{chrII}$.

Yes, thank you for reporting this mistake. We changed the karyotype as $2n + 2 \cdot \text{chrII}$.

L535-547. The AIS genome (Fig 5E) is very interesting. Does it contain complete rDNA units from both Sk and Sc?

The two RDN from *S. cerevisiae* and *S. kudriavzevii* (CR85) are highly similar but it is possible to discriminate them because one is 99% identical to AIS while the other is 98%. We used fasta36 to double-check what we initially reported using wgVista and found an organization identical to the one reported in Fig. 4E. Therefore, there are both rDNA sequences in the assemblies and these should be complete. However, the initial RDN region of the HP2 haplotype remains undefined, both sequences being ~97-98% which could mean that the two units would be mixed up (as symbolized by the grey box in the figure).

L648. Change to "clade 13 (lab-related) in which..."

Done

L671. I'm not sure what "that would" means in the sentence beginning "A genomic clock that would synchronize...". The sentence seems to be grammatically incorrect, and I'm not sure if this clock is a hypothetical idea or a real observation. Maybe change it to "synchronizes" or "has been postulated to synchronize" ?

We agree that this sentence was incorrect. We change it to "It has been proposed that a genomic clock would coordinate the pace of fixation between amino acid substitutions and large-scale rearrangements in bacteria and yeast".

Signed review by Ken Wolfe

Reviewer #2:

Remarks to the Author:

The manuscript by O'Donnell et al. describes complete genome assemblies for 142 representative strains of the model yeast *Saccharomyces cerevisiae*. Using these assemblies, the authors characterize the structural variation of the species in great detail.

The analyses are generally well done and clearly described, although the (very broad) scope of the data and the in-depth analyses means that there are a few places where it was hard for me to figure out what the authors did or meant (at least I couldn't after 2-3 reads). Here are a few cases:

Lines 97-98 and Table 1: it is not clear to me why the authors do not provide the haplotype phased assemblies for the 13 polyploid strains. As far as I understand, the authors provide 13 genome assemblies, one for each strain, rather than 46 assemblies (6*3 for the 6 triploid strains and 7*4 for the 7 tetraploid strains)

We apologize for not being clear on this matter. We have added a new panel on Supp Fig. 1 (panel B) to illustrate the relationships between the 142 strains and the 197 nuclear genome assemblies. To answer more specifically to the reviewer's comment on polyploids, we generated one collapsed and one phased assembly (haplotypes were not independently assembled) per heterozygous polyploid genome. We think that the new figure panel is now clearly illustrating this.

Lines 99-101: Not clear why only 135 mitochondrial genomes are available from 142 genomes. Can you briefly add an explanation?

We thank the reviewer for this comment. We thoroughly rechecked all publicly available assemblies and found that the Y55 genome assembly included a complete mitochondrial contig that we omitted to include. We have annotated and added this assembly to the ScRAP, totaling now 136 mitochondrial genomes. We have also added an explanation in the Supplementary Information to describe why only 136/142 strains have a mitochondrial assembly:

"The ScRAP contains only 136 mitochondrial assemblies because (i) 3 strains ALH_1c, ASB and ASG) are petite mutants devoid of mitochondrial DNA, (ii) 2 strains have highly incomplete mitochondrial assemblies probably due to low coverage (BDH and RM11) and (iii) the BY4742 public genome assembly does not include a mitochondrial contig."

Section beginning on line 328: what are SNVs and how they differ from SNPs? (they are never defined - I presume they both refer to single nucleotide polymorphisms)

Lines 344-346: The authors infer that there is lower information / signal on the SV tree than on the SNP tree but then conclude that the signal from the two types of data is complementary. I am not sure I understand how could this be, especially if one considers a specific genomic region that contains both SNPs and SVs (I would expect that patterns of variation in both types of data would reflect the same evolutionary history).

We agree with the reviewer that it is not completely straightforward to explain why SVs and SNVs would reflect different evolutionary histories but SVs are more parsimonious mutational events and don't necessarily share the same selective coefficient as SNVs. In addition, SVs have very low MAF and could be less prone to admixture. Therefore, they might in some instances follow more faithfully a vertical inheritance than SNVs.

In a few places, the authors use over-the-top language that should be toned down. Here are a few cases:

Lines 102-104: Providing a brief summary of the contiguity and completeness data would have been better than telling us that the dataset is "unprecedented"

We changed the corresponding sentence into "Genomic metrics reveal high contiguity (16 to 17 scaffolds for haploid/collapsed assemblies) and completeness levels (all 32 telomeres assembled in

51 haploid/collapsed assemblies) across the ScRAP assemblies ”

Various lines: "first rigorous" (line 77), "unprecedented" (line 102), "notable improvement" (line 115), etc. In my opinion these should be all removed and the authors should let the data speak for themselves...

We removed all instances of such terms.

Line 126: the title of this section is "SVs impact gene expression" - given the subtle effect, it would be better to say that "SVs can impact gene expression" since only a small fraction of SVs are shown to affect gene activity.

This has been changes accordingly

I believe that all these points are straightforward to address and that the dataset provided will be of high value to the community. If I had one major concern with the manuscript, this would be the lack of a clear take-home message or conclusion. I understand that the authors have characterized structural variation of a yeast species in unprecedented detail, but what does this tell us? I see lots of detailed characterizations of the different types of events that can occur and the frequencies with which they occur, but I am missing the take-home message. Tellingly, there are ~20+ pages of results and only 1 page of discussion. Maybe that's a whole separate study, but I think the authors are missing here a great opportunity to state the broader relevance of their results and findings for population biology and genome evolution.

We thank the reviewer for this comment and fully agree on the importance of a clear take-home message. We have therefore largely rewritten the summary paragraph in order to make the important points of this work, which were also mentioned by reviewer 1, appear more clearly.

We deliberately aimed to be concise with the discussion as the length of the paper already exceeds the limit of the journal formatting guidelines . We have further focused on the SVs as the central core of this work and their crucial role in globally shaping the species genome evolution.

Minor points:

Line 567: "in less than" -> "in fewer than"

Done

Line 268: "majoritarily" -> "mostly"

Done

Lines 670-673: check this sentence and rephrase - it doesn't quite make sense

This sentence was rephrased as follows:

“It has been proposed that a genomic clock would coordinate the pace of fixation between amino acid substitutions and large-scale rearrangements in bacteria and yeast^{51,52}. However, this clock seems to tick at a different pace depending on the ploidy and zygosity of the genome.”

Reviewer #3:

Remarks to the Author:

The authors generate the *Saccharomyces cerevisiae* Reference Assembly Panel (ScRAP) comprising telomere-to-telomere genome assemblies for 142 strains from various geographical and ecological distributions. The ScRAP is important for population genomics studies of *S. cerevisiae*, in terms of phylogeny, ploidy, heterozygosity and ecological origins. Using the generated ScRAP dataset, the authors characterize the structural variant (SV) and reveal valuable findings regarding the evolutionary dynamics of complex regions. The high quality of data, the precise presentations, as well as the appropriate usage of statistics, are all satisfactory. The conclusions are reliable. The

manuscript is also well written. In general, I suggest to accept this manuscript.

There are only some minor issues:

1. In Page 5, L98 states 13 strains are heterozygous polyploidy, which should include 6 triploid and 7 tetraploid as shown in Table 1. However, the number of triploid is 7 in figure 1A and 1C. Please check and confirm the information.

The difference is due to the presence of 1 homozygous triploid, as explained in the legend of Figure 1 and in Supp Fig. 1. So, there is no problem with the numbers, there are actually 14 polyploids but only 13 heterozygous polyploids.

2. It is interesting that all of the triploid and tetraploid (13 or 14 strains in total) in this study are isolated from fermentations, especially from wine and beer. I wonder if the authors find anything new in the structure variants analysis that may help to explain the connection between multi-ploidy and fermentation ability (or special food flavor)?

It is true that all polyploids cluster in clades related to fermentation. However, there is one triploid strain CRE that was isolated from a termite mound in Ecuador. This strain phylogenetically clusters into the Asian fermentation clade but was isolated from the wild. Unfortunately, we did not find any specific variant that could explain some fermentative or organoleptic properties of the polyploid strains so far, but such association between polyploidy and adaptation to fermentation environment is definitely interesting to explore with future studies.

Decision Letter, first revision:

24th Apr 2023

Dear Dr. Fischer,

Thank you for submitting your revised manuscript "142 telomere-to-telomere assemblies reveal the genome structural landscape in *Saccharomyces cerevisiae*" (NG-A61280R). It has now been seen by the original referees and their comments are below. The reviewers find that the paper has improved in revision, and therefore we'll be happy in principle to publish it in Nature Genetics, pending minor revisions to comply with our editorial and formatting guidelines.

Sincerely,
Wei

Wei Li, PhD
Senior Editor
Nature Genetics
New York, NY 10004, USA

www.nature.com/ng

Reviewer #1 (Remarks to the Author):

I'm happy with the authors' response to my comments, and the changes they have made to the text and figures. I'm still very enthusiastic about this manuscript. I particularly appreciate the extra columns the authors have added to the list of SVs in Table S5, showing for each SV its allele frequency and the names of strains that contain it, which will make the SV data easily accessible to other future users.

Reviewer #2 (Remarks to the Author):

The authors have satisfactorily addressed my comments.

Reviewer #3 (Remarks to the Author):

All comments/concern were addressed.

Final Decision Letter:

26th Jun 2023

Dear Dr. Fischer,

I am delighted to say that your manuscript "Telomere-to-telomere assemblies of 142 strains characterize the genome structural landscape in *Saccharomyces cerevisiae*" has been accepted for publication in an upcoming issue of Nature Genetics.

Your paper will be published online after we receive your corrections and will appear in print in the next available issue. You can find out your date of online publication by contacting the Nature Press Office (press@nature.com) after sending your e-proof corrections. Now is the time to inform your Public Relations or Press Office about your paper, as they might be interested in promoting its publication. This will allow them time to prepare an accurate and satisfactory press release. Include your manuscript tracking number (NG-A61280R1) and the name of the journal, which they will need when they contact our Press Office.

Please note that *Nature Genetics* is a Transformative Journal (TJ). Authors may publish their research with us through the traditional subscription access route or make their paper immediately open access through payment of an article-processing charge (APC). Authors will not be required to make a final decision about access to their article until it has been accepted. [Find out more about Transformative Journals](https://www.springernature.com/gp/open-research/transformative-journals)

Authors may need to take specific actions to achieve [compliance](https://www.springernature.com/gp/open-research/funding/policy-compliance-faqs) with funder and institutional open access mandates. If your research is supported by a funder that requires immediate open access (e.g. according to [Plan S principles](https://www.springernature.com/gp/open-research/plan-s-compliance)) then you should select the gold OA route, and we will direct you to the compliant route where possible. For authors selecting the subscription publication route, the journal's standard licensing terms will need to be accepted, including [self-archiving-and-license-to-publish](https://www.nature.com/nature-portfolio/editorial-policies/self-archiving-and-license-to-publish). Those licensing terms will supersede any other terms that the author or any third party may assert apply to any version of the manuscript.

Please note that Nature Portfolio offers an immediate open access option only for papers that were first submitted after 1 January, 2021.

To assist our authors in disseminating their research to the broader community, our SharedIt initiative provides you with a unique shareable link that will allow anyone (with or without a subscription) to

read the published article. Recipients of the link with a subscription will also be able to download and print the PDF.

If you have not already done so, we invite you to upload the step-by-step protocols used in this manuscript to the Protocols Exchange, part of our on-line web resource, natureprotocols.com. If you complete the upload by the time you receive your manuscript proofs, we can insert links in your article that lead directly to the protocol details. Your protocol will be made freely available upon publication of your paper. By participating in natureprotocols.com, you are enabling researchers to more readily reproduce or adapt the methodology you use. [Natureprotocols.com](http://natureprotocols.com) is fully searchable, providing your protocols and paper with increased utility and visibility. Please submit your protocol to <https://protocolexchange.researchsquare.com/>. After entering your nature.com username and password you will need to enter your manuscript number (NG-A61280R1). Further information can be found at <https://www.nature.com/nature-portfolio/editorial-policies/reporting-standards#protocols>

Sincerely,
Wei

Wei Li, PhD
Senior Editor
Nature Genetics
New York, NY 10004, USA
www.nature.com/ng